# Application of the Townsend-George theory for free shear flows to single and double wind turbine wakes - a wind tunnel study

Ingrid Neunaber[1,2], Joachim Peinke[2], and Martin Obligado[3]

[1]LHEEA – École Centrale de Nantes, CNRS, 1 Rue de la Noë, 44321 Nantes, France
[2]Institute of Physics and ForWind, University of Oldenburg, Küpkersweg , 26129 Oldenburg, Germany
[3]Université Grenoble Alpes, CNRS, Grenoble-INP, LEGI, F-38000, Grenoble, France

**Correspondence:** Joachim Peinke (peinke@uol.de); Ingrid Neunaber (ingrid.neunaber@uol.de)

**Abstract.** The evolution of the mean velocity and the turbulence downstream of wind turbine wakes within the atmospheric boundary layer has been studied over the past decades but an analytical description is still missing. One possibility to improve the comprehension is to look into the modeling of turbulent bluff body wakes. There, by means of the streamwise scaling of the centerline mean velocity deficit, the nature of the turbulence inside a wake can be classified. In this paper, we introduce the analytical model from classical wake theory as introduced by A. Townsend and W.K. George. To test the theories, data was obtained from wind tunnel experiments using hot-wire anemometry in the wakes of a single model wind turbine and a model wind turbine operating in the wake of an upstream model wind turbine. First, we test whether the requirements under which the Townsend-George theory is valid are fulfilled in the wake of a wind turbine. Based on this verification we apply the Townsend-George theory. Further, this framework allows to distinguish between two types of turbulence, namely equilibrium and non-equilibrium turbulence. We find that the turbulence at the centerline is equilibrium turbulence and that non-equilibrium turbulence may be present at outer parts of the wake. Finally, we apply the Townsend-George theory to characterize the wind turbine wake, and we compare the results to the Jensen and the Bastankhah-Porté-Agel models. We find that the recent developments from the classical bluff body wake formalism can be used to further improve the wind turbine wake models. Particularly, the classical bluff body wake models perform better than the wind turbine wake models due to the presence of a virtual origin in the scalings, and we demonstrate the possibility of improving the wind turbine wake models by implementing this parameter. We also see how the dissipation changes across the wake, which is important to model wakes within wind farms correctly.

## 1 Introduction

Wind turbines are usually clustered in wind farms with the consequence that downstream turbines operate depending on the wind direction and wind speed in the turbulent wakes of upstream turbines (e.g. Barthelmie et al. (2007); Sun et al. (2020)). These wakes are characterized above all by means of the mean velocity deficit that decreases in the far wake with increasing distance from the turbine. Since this velocity deficit leads to power losses of downstream turbines, knowledge of the recovery of the wind velocity is important. Therefore, over the past decades, several empirical and engineering wake models have been derived to describe a single turbine's velocity deficit, see for instance the reviews of Porté-Agel et al. (2020) and Göçmen et al.

(2016). All these models find application in the layout optimisation and, once the wind farm is built, in the control mechanisms. Ideally, these models have to be simple, fast, and computationally inexpensive despite the complex flow configuration. In addition, a model describing an axisymmetric turbulent wind turbine wake from only a few fundamental and robust assumptions is still lacking. On the contrary, the axisymmetric turbulent wake of a bluff body is a canonical turbulent free shear flow which has been studied intensively during the last decades and which can be modeled by means of an analytical model. The modelling of

these flows relies on the theories proposed by Townsend (1976) and George (1989), even though results have been inconclusive and even sometimes contradictory (Johansson et al. (2003)). While the wake of a wind turbine does exceed the complexity of historical wake investigations of bluff bodies due to the turbine's rotation and active interaction with the background flow, the question arises to what extent classical wake models can be applied. So far, only the power law relationship between the velocity deficit and the streamwise distance that originates from classical wake theory has been used in empirical wind turbine

wake models to predict the centerline velocity recovery (cf. Porté-Agel et al. (2020)).

Within the framework of turbulence, the prediction of the centerline mean velocity deficit $\Delta U$ and the wake width $\delta$ with increasing streamwise distance $X$ of the wake generating model are maybe the most fundamental and relevant problems that need to be solved (e.g. George (1989)) [1]. Similarly to the standard engineering models derived for wind turbine wakes, the analytical model for the evolution of bluff body wakes can only be derived for the far wake. The far wake is typically identified as

the part of the wake where the shear layers that evolve between the faster ambient flow and the lee of the object of investigation have met and the turbulence is fully developed. Thus, the Townsend-George theory for free shear flows allows to predict the downstream evolution of $\Delta U$ and $\delta$ for the axisymmetric, boundary-free, turbulent wake of a bluff body sufficiently far downstream. This analytical model is based on two cornerstone assumptions regarding the turbulence evolution. The first one is the self-similar behavior of the one-point turbulence statistics. The second cornerstone assumption focuses on the energy cascade:

In the Richardson-Kolmogorov phenomenology, the energy is injected on the large scales, i.e. large vortex structures, and when these vortices decay, the energy is transferred towards smaller scales, and therefore the inter-scale energy transfer is balanced by the turbulent dissipation rate $\varepsilon$. This cascade is therefore in equilibrium, i.e. the energy that is injected into the system on large scales is transferred completely to smaller and smaller scales until it dissipates, while the power spectral density shows the famous $E(f) \propto f^{-5/3}$ decay in the inertial sub-range (where $f$ is the frequency in Hz). In this case, the dissipation rate

scales as $\varepsilon = C_\varepsilon K_c^{2/3}/L$ along the centerline with $C_\varepsilon = const.$ at high Reynolds numbers[2]. $K_c$ denotes the centerline turbulent kinetic energy and $L$ denotes an integral length scale describing the energy-containing structures. $C_\varepsilon = const.$ is therefore one of the most critical points in the theory. The scaling of the dissipation rate $\varepsilon$ is a scaling used in many aspects of turbulence theory and in the modelling and understanding of many turbulent flows (see for example Tennekes and Lumley (1972)). As better detailed in section 2, the closure provoked by this assumption leads to the standard equilibrium streamwise scalings for

$\Delta U$ and $\delta$.

Yet, in the last years, evidence has been found that some axisymmetric turbulent wakes do not follow the standard scalings

---

[1]Note that a fundamental difference between the engineering wake models and the analytical bluff body wake model is the modeling of the wake width: In wind turbine models, a growth rate is derived empirically, while in the bluff body wake analysis, an analytical model for the wake width is derived.

[2]$C_\varepsilon$ may change for different flows but for a certain set of boundary conditions, $C_\varepsilon = const.$ independently of the Reynolds number

of $\Delta U$ and $\delta$. Different experimental studies and direct numerical simulations of bluff plates with both regular and irregular peripheries found that $C_\varepsilon$ is not constant, but instead it goes as $C_\varepsilon \sim Re_G^m / Re_L^n$ (e.g. Vassilicos (2015); Dairay et al. (2015); Obligado et al. (2016); Nedic et al. (2013); Es-Sahli et al. (2020)). $Re_G$ is a Reynolds number that depends on the inlet con-

ditions, and $Re_L$ is a local, streamwise, position-dependent one (Vassilicos (2015)). The exponents $n$ and $m$ have been found to be very close to unity; $m = n = 1$ for large values of the Taylor Reynolds number $Re_\lambda$ (cf. equation A4). For an axisymmetric wake, $Re_G = \sqrt{\Lambda} U_\infty / \nu$, with $\Lambda$ the frontal area of the plate, $U_\infty$ the inlet velocity and $\nu$ the kinematic viscosity of the flow. The local Reynolds number is defined as $Re_L = \delta u' / \nu$, with $u'$ the RMS value of the fluctuating streamwise velocity. Within these definitions, the non-equilibrium dissipation scaling can also be written as $C_\varepsilon \sim Re_G^{1/2} / Re_\lambda$. These anomalous

non-equilibrium scalings of $\Delta U$ and $\delta$ have been reported for experiments at $X/\sqrt{\Lambda} < 50$, therefore a range of interest within some wind energy applications.

This illustrates that an investigation of $C_\varepsilon$ in the wake gives us a new tool to classify the turbulence in the wake with $C_\varepsilon = const.$ being an indication of "standard" equilibrium turbulence with properties of homogeneous isotropic turbulence, and $C_\varepsilon \sim Re_G^m / Re_L^n$ (or equivalently $C_\varepsilon \sim Re_G^{1/2} / Re_\lambda$) indicating non-equilibrium turbulence with its own set of properties.

The presence of a different dissipation scaling of $\varepsilon$ within wind turbine wakes would have important consequences regarding modelling of wind farms and numerical simulations: Many different numerical models implicitly assume the standard dissipation scalings (for more details, see Vassilicos (2015)). However, a different dissipation scaling also implies different streamwise scalings for $\Delta U$ and $\delta$ (see section 2). The disregard of this point shows that we have not yet fully understood the physics underlying the simplest possible configuration in wind energy: a single turbine facing a uniform, stationary, laminar

flow. Therefore, we follow a different path in this paper than the one engineering models are based on, namely the conservation of mass and momentum to describe the mean quantities. The wake of a wind turbine can only be fully understood if the turbulence evolution is properly characterised, and the presented approach can help to integrate turbulent fluctuations in the wake. Furthermore, the fact that different dissipation scalings (the dissipation being a small-scale parameter) result in different spreading properties of the wake (a large-scale parameter) shows that the turbulence cascade controls how the wake

is evolving. Therefore, the initial conditions, such as the wake generator and the background flow, become important as they ultimately control the evolution of the cascade. It is thus important to study how the dissipation scalings are affected by the inflow conditions.

In contrast to the axisymmetric turbulent wake of a bluff body, no systematic work on the turbulence decay and on the energy cascade has been made yet in the turbulent wind turbine wake. However, Okulov et al. (2015) applied the standard equilib-

rium bluff body wake scalings derived from the Richardson-Kolmogorov phenomenology to experimental data to justify the presence of power-law decays for $\Delta U$ and $\delta$. In addition, Stein and Kaltenbach (2019) performed a systematic study on the nature of the dissipation scaling, testing also the non-equilibrium scaling, but without measuring nor taking into account $C_\varepsilon$ in the discussion. Furthermore, for the latter, the turbine studied was within a turbulent boundary layer background flow which means that the turbulence evolution was additionally influenced by the turbulent background flow.

In this paper, we present an experimental study on the axisymmetric turbulent wake generated by a wind turbine in a wind tunnel via hot-wire anemometry. We study the wake produced by a single turbine in uniform laminar inflow. Furthermore, we also

analyse the behaviour of a turbine within a turbulent background: this is achieved by placing a second turbine downstream of the first one. The second turbine is tested at two different radial positions, that allow us to explore the behaviour of a turbulent wake within two different, relevant configurations within a wind farm. As this work serves as a proof of the applicability of the Townsend-George theory, we do not include an investigation of the influence of an atmospheric boundary layer (ABL) profile where inflow characteristics may differ. However, it is generally assumed that the wake of a wind turbine in an ABL can be seen as a superposition between the ABL profile and the mean velocity deficit (cf. Bastankhah and Porté-Agel (2014)). This is in agreement with Neunaber et al. (2021) where the Townsend-George theory also gives good results in the case of field measurements obtained in the wake of a full-scale turbine using a LiDAR.

We perform a systematic study on the applicability of the Townsend-George theory to wind-turbine-generated wakes for the different incoming flows. For this, we explore the streamwise range in which the requirements are fulfilled. We also study the behaviour of $C_\varepsilon$ for this flow, i.e. whether $C_\varepsilon = const.$, and therefore of the scalings of $\Delta U$ and $\delta$. By doing so, we identify the nature of the energy cascade for the different wind turbine configurations. We find that all three turbine configurations are in good agreement with the Richardson Kolmogorov energy cascade. Furthermore, we verify that, to some extent, they also are within the conditions and hypotheses required to apply the Townsend-George theory.

In the following, we will introduce the Townsend-George theory in section 2. Here, the framework as well as the necessary requirements to apply the theory are introduced. Also, two important engineering wake models are introduced for a comparison. Then, the setup is presented in section 3.1, and the fulfillment of the requirements is investigated in section 3.2. The theory is applied in section 4 and the results are discussed in section 5. This paper ends with a conclusion in section 6.

## 2 Theory

### 2.1 Bluff body wakes

Bluff body wakes have been the object of intensive analyses for the past 60 years, and similarly to the study of wind turbine wakes, the downstream evolution of the centerline mean velocity deficit $\Delta U$ and the wake width $\delta$ are of interest.

In this work, we will follow the revisited Townsend-George theory (Dairay et al. (2015)). The classical equilibrium predictions and the non-equilibrium predictions rely on the axisymmetry of turbulence wake statistics, the self-preservation of $(U_\infty - U(X,Y))/\Delta U$ (with $U(X,Y)$ the streamwise mean velocity, $Y$ the span-wise coordinate, $\Delta U = U_\infty - U_c$ the centerline velocity deficit and $U_c$ the streamwise centerline mean velocity), the turbulent kinetic energy $K$, the Reynolds shear stress $R_{xr}$, the turbulence dissipation $\varepsilon$, and on a scaling law for the centerline turbulence dissipation that is determined by the behavior of $C_\varepsilon$ due to $\varepsilon = C_\varepsilon K_c^{2/3}/L$. Both sets of predictions are obtained from the Reynolds-averaged streamwise momentum and turbulent kinetic energy equations leading to a closed set of equations for $\Delta U(X)$ and the wake width $\delta(X)$.

These hypotheses also allow to express the momentum conservation within the wake as $(\theta/\delta)^2 = \Delta U/U_\infty$. The momentum thickness $\theta$ is defined by $\theta^2 = \frac{1}{U_\infty^2} \int_0^\infty U (U_\infty - U) r \mathrm{d}r$ which is constant with $X$, and the wake's width is characterised here by

the integral wake width $\delta$ defined by $\delta^2(X) = \frac{1}{\Delta U} \int_0^\infty (U_\infty - U) \, r \, \mathrm{d}r$ (both quantities are linked via momentum conservation as $(\theta/\delta)^2 = \Delta U/U_\infty$).

The equilibrium predictions for axisymmetric turbulent wakes (see Townsend, 1976; George, 1989) for the streamwise evolution (along $X$) of $\Delta U$ and $\delta$ are

$$\Delta U(X) = A_{EQ} U_\infty \left( (X - X_{0,EQ})/\theta \right)^{-2/3}, \tag{1}$$

$$\delta(X) = B_{EQ} \theta \left( (X - X_{0,EQ})/\theta \right)^{1/3}, \tag{2}$$

where $A$ and $B$ are dimensionless constants, and $X_0$ a virtual origin (that appears naturally from the mathematical formulation and has to be the same for $\Delta U$ and $\delta$).

In contrast, the non-equilibrium predictions are

$$\Delta U(X) = A_{NEQ} U_\infty \left( (X - X_{0,NEQ})/\theta \right)^{-1}, \tag{3}$$

$$\delta(X) = B_{NEQ} \theta \left( (X - X_{0,NEQ})/\theta \right)^{1/2}. \tag{4}$$

The only difference between equilibrium and high-Reynolds number non-equilibrium scalings for $\Delta U$ and $\delta$ is in the scaling of the centerline value of $\varepsilon$: As discussed in the introduction, the non-equilibrium turbulence dissipation law is $\varepsilon_c = C_\varepsilon K_c^{3/2}/\delta$ where $K_c$ is centerline turbulent kinetic energy, $\varepsilon_c$ is the centerline turbulence dissipation and $C_\varepsilon$ is a dimensionless coefficient which is constant in equilibrium turbulence but proportional to the ratio of two Reynolds numbers, $Re_G^m/Re_L^n$, in the case of non-equilibrium turbulence.

Another important assumption used throughout the theory is that the large scales of the turbulence are represented by the wake width $\delta$, and therefore $\delta \propto L$ (Cafiero et al. (2020)). This is an important, and frequently overlooked assumption within the Townsend-George theory. To the authors' knowledge, it has not been verified for wind turbine wakes.

To summarize, the Townsend-George theory of axisymmetric turbulent wakes relies on the following flow properties:

**Requirements to apply the Townsend-George theory.**

1. The flow has to be decaying turbulence: we will focus on the range for which the turbulent kinetic energy is a decreasing function of the streamwise distance.

2. The flow has to be turbulent. A way to verify this requirement is to check that the velocity signal has
   a) a large value of $Re_\lambda$ (i.e. $Re_\lambda > 200$), and
   b) a power spectral density with an inertial sub-range that decays according to the $E(f) \propto f^{-5/3}$ power law.

3. The mean streamwise velocity and the turbulence quantities discussed above also have to be self-preserving and thus show a self-similar scaling (for a more detailed study on this point, we refer the reader to Dairay et al. (2015)).

4. The mean velocity components and the turbulence quantities have to be axisymmetric.

5. As stated, the flow has to be in a streamwise range for which the longitudinal integral length scale is proportional to the wake width ($L \propto \delta$).

If all conditions are fulfilled, the theory predicts power-law decays for the velocity deficit and the wake width. Furthermore, the exponents of such power-laws can be related to the dissipation scaling of $\varepsilon$ in the flow.

In order to disentangle the different dissipation scalings and thus identify equilibrium and non-equilibrium turbulence, two criteria have to be checked:

**Criteria for Equilibrium/Non-Equilibrium Turbulence.**

i Does $C_\varepsilon = const.$ hold? In this case, no Reynolds number dependence of $C_\varepsilon$ should be seen.

*yes*: equilibrium turbulence

*no* : indication for non-equilibrium turbulence

ii The Taylor Reynolds number $Re_\lambda$ and the local Reynolds number $Re_L$ need to change with downstream distance in order to verify criterion i. More specifically, $Re_L$ has to decrease according to George (1989) in the case of non-equilibrium turbulence.

If $Re_L$ and $Re_\lambda$ do not change, it is therefore not possible to draw conclusions on the occurrence of equilibrium and non-equilibrium turbulence and the results are inconclusive.

In the following, we will refer to the equilibrium scaling as EQ scaling, and the non-equilibrium scaling as NEQ scaling.

## 2.2 Wind turbine wake models

In the past, a lot of effort was put into the understanding and finding a description of the wake flow of wind turbines, especially its recovery on the centerline and the wake expansion, by carrying out experiments and simulations. Additionally, different theories have been underlain. However, an analytical model that is capable of including different inflow conditions and turbine operations does still not exist. In the following, we will briefly introduce two commonly used wake models that we will compare to the EQ and NEQ wake models.

The first model we discuss, developed by N. O. Jensen in 1983, is based on the conservation of momentum (cf. Jensen (1983)). It is assumed that the normalized velocity deficit

$$\frac{\Delta U}{U_\infty} = \frac{U_\infty - U_W}{U_\infty} \tag{5}$$

with the wake velocity $U_W$ is top-hat shaped and evolves with increasing downstream distance $X$ according to

$$\frac{\Delta U}{U_\infty} = \left(1 - \sqrt{1 - c_T}\right) \cdot \left(1 + \frac{2k_J X}{D}\right)^{-2}. \tag{6}$$

$U_\infty$ denotes the unperturbed inflow velocity, $U_W$ the streamwise wake velocity, $c_T$ the turbine's thrust coefficient, $D$ the turbine's diameter and $k_J$ the wake expansion, that is assumed to be constant. While Jensen suggests $k_J = 0.1$, $k_J = 0.075$ is

suggested in Barthelmie et al. (2006).

A more recent model that includes the conservation of mass in addition to the conservation of momentum is the one derived by Bastankhah and Porté-Agel (2014). Here, the recovery of the centerline velocity deficit is combined with a Gaussian velocity profile,

$$\frac{\Delta U}{U_\infty} = \underbrace{\left(1 - \sqrt{1 - \frac{c_T}{8\left(k_{BP} \cdot X/D + 0.2\sqrt{\beta}\right)^2}}\right)}_{\text{centerline velocity deficit}} \cdot \underbrace{\exp\left(-\frac{((Z - Z_h)/D)^2 + (Y/D)^2}{2\left(k_{BP} \cdot X/D + 0.2\sqrt{\beta}\right)^2}\right)}_{\text{Gaussian velocity profile}}. \qquad (7)$$

In this formula, $Y$ and $Z$ denote the span-wise and wall-normal coordinates, $Z_h$ denotes the hub height, and $k_{BP}$ is the wake growth rate that needs to be specified and is in the order of magnitude of $0.03 \pm 0.02$ (Bastankhah and Porté-Agel (2014), Okulov et al. (2015)). $\beta$ is given by the equation

$$\beta = \frac{1 + \sqrt{1 - c_T}}{2\sqrt{1 - c_T}}. \qquad (8)$$

In the following, we will call this wake model the BP model.

It can be seen that the two approaches that were presented above to describe the wind turbine wake differ significantly from the classical wake models that were derived from the perspective of turbulence.

## 3   Experimental setup & verification of requirements

### 3.1   Setup

In the following, the experimental setup used to carry out the experiments evaluated in this study is introduced. The experiments have been performed in Oldenburg's Large Wind Tunnel (OLWiT) at the University of Oldenburg that has an inlet of $(3 \times 3)\,\text{m}^2$ and a closed test section of 30m length (cf. Kröger et al. (2018)). In the empty test section, the background turbulence intensity of the flow is $TI = 0.3\%$ (where $TI = u'/U$), and velocities of up to $42\,\text{ms}^{-1}$ can be reached. A closed-loop control keeps the velocity constant, and to correct for changes, the flow temperature, the ambient pressure, and the humidity are constantly monitored. During all experiments presented here, the uniform inlet velocity was $U_\infty = (7.55 \pm 0.05)\,\text{ms}^{-1}$.

Two three-bladed horizontal axis model wind turbines of the same type have been used in the experiments (cf. Schottler et al. (2018)). In the following, they will be referred to as turbine 1 and turbine 2. Both turbines have a rotor diameter of $D = 58\,\text{cm}$ and a hub height of $h = 72\,\text{cm}$ (resulting in an inlet Reynolds number of $Re_G = 3 \times 10^5$ based on the square root of the rotor surface). The rotor load is controlled using a closed-loop control to ensure a performance of the turbine at the optimal tip speed ratio of $TSR \approx 5.8$ in the full-load range (cf. Petrović et al. (2018); Neunaber (2019)). This is achieved be measuring the rotor torque and adapting it to achieve the optimal rotational speed that can be taken from a look-up table. The thrust coefficient of the whole turbine was measured by placing the turbines on a force balance, and it is $c_{T,t1} \approx 1$ for turbine 1 and $c_{T,t2} \approx 1.07$ for turbine 2 (see Neunaber (2019)). The thrust coefficient of the tower and the nacelle was also measured and it yields $c_{T,n} \approx 0.12$

so that the turbine operates close to the ideal thrust coefficient of $c_{T,id} = 8/9$ derived from the Betz limit.

As shown in figure 1, an array of six 1d hot-wire probes with sensor lengths of $1.25\,$mm and a sensor diameter of $5\,\mu$m was used to measure the wake downstream of three different wind turbine array configurations at the hub height with a very high downstream resolution of $\Delta X/D = 0.17$. The data was collected using a StreamLine 9091N0102 frame with 91C10 CTA Modules. A temperature correction, as proposed by Hultmark and Smits (2010), was applied to the data. At each position, $1.2 \cdot 10^6$ data points were collected with a sampling frequency of $f_s = 15\,$kHz. A hardware low-pass filter with a cut-off frequency of $10\,$kHz has been used.

Three wake configurations were investigated (cf. figure 1):

1. The wake of turbine 1 exposed to the uniform, laminar inflow in the range between $X/D = 0.55$ and $X/D = 12.62$. This configuration will be referred to as *turbine 1*. Based on the rotor, the blockage of the wind tunnel is 3% in this configuration.

2. The wake of turbine 2 positioned $5.17D$ downstream in the wake of turbine 1 between $X/D = 0.55$ and $X/D = 8.66$ with respect to turbine 2[3]

   (a) First, turbine 2 is exposed to the wake of turbine 1. Both are aligned, and therefore share the centerline axis. As shown in Neunaber (2019), the inflow is thus turbulent with an average inflow velocity of $3.4\,$ms$^{-1}$ ($2.1\,$ms$^{-1}$ in the center, $5.5\,$ms$^{-1}$ at the blade tip). This configuration will be referred to as *turbine 2 mid*.

   (b) Then, turbine 2 is moved $0.5D$ to the side so that it is exposed to an inhomogeneous half-wake inflow. As shown in Neunaber et al. (2020), the inflow is thus turbulent and intermittent. The average inflow velocity is $5.1\,$ms$^{-1}$ with a gradient from $2.1\,$ms$^{-1}$ at one side of the rotor to $7.4\,$ms$^{-1}$ at the other side of the rotor. This configuration will be referred to as *turbine 2 side*.

In the cases of turbine 1 and turbine 2 mid, we measure one half of the wake because the wake is expected to be axisymmetric in uniform inflow (Stevens and Meneveau (2017)). The horizontal position of the hot-wire array is not changed in the case of turbine 2 side so that the full wake is measured in this case.

## 3.2 Verification of the Requirements

As explained in section 2, certain requirements have to be fulfilled to justify the use of the Townsend-George theory. Therefore, in the following, we will verify that the listed criteria are accomplished.

### 3.2.1 Turbulence intensity

In figure 2, the centerline evolution of the turbulence intensity is plotted for the three inflow scenarios. As also discussed in Neunaber et al. (2020), the turbulence intensity first decreases in the nacelle's lee, then increases when the turbulence builds up due to the expansion of the shear layers evolving between the faster ambient flow and the slower wake and afterwards decreases

---

[3]This is equal to $X/D = 13.82$ with respect to turbine 1.

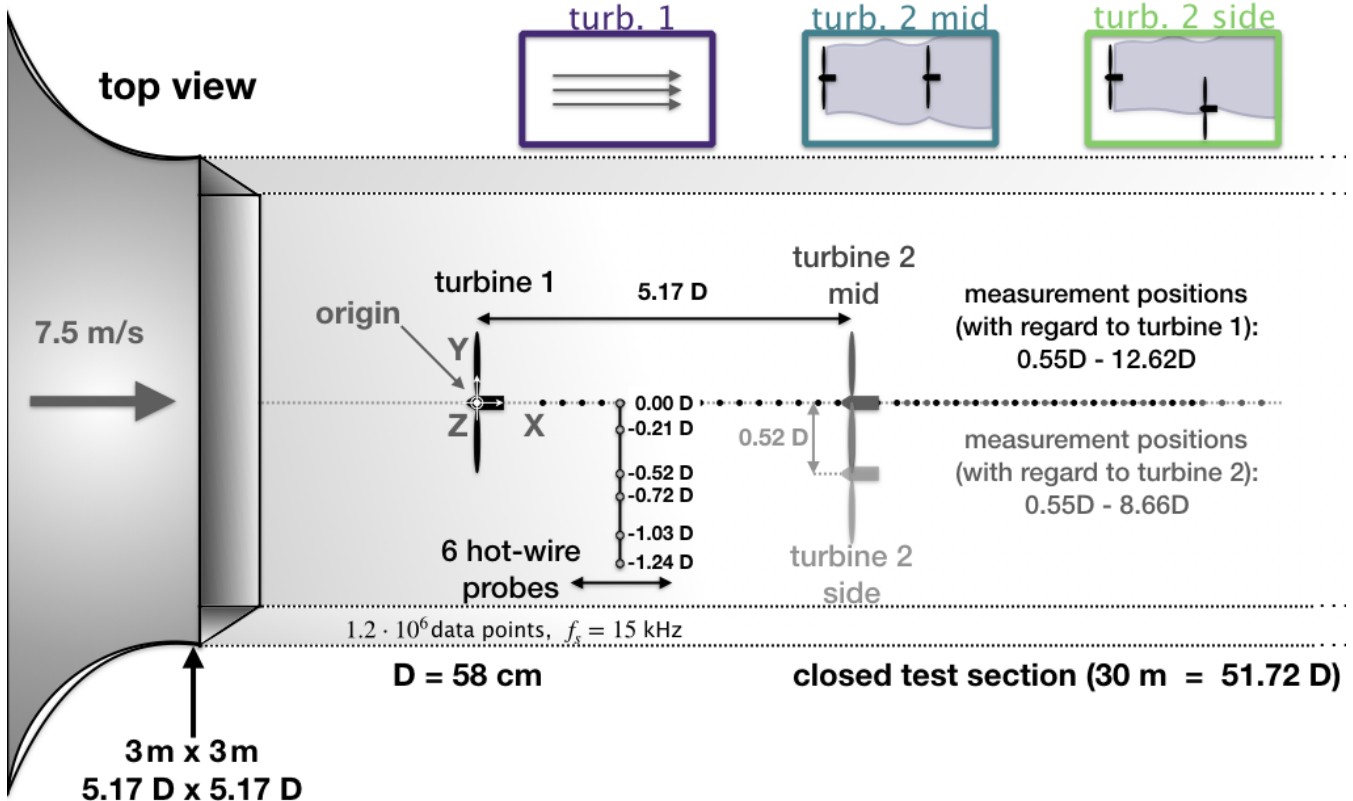

**Figure 1.** Setup (modified from Neunaber (2019)): A hot-wire array is used to measure the wakes of 1. a single turbine (plot color: dark purple), 2. a turbine exposed to the wake of an upstream turbine (plot color: turquoise), and 3. the half-wake of an upstream turbine (plot color: green).

in the far wake where the turbulence decays. The first requirement that has to be fulfilled in order to apply the Townsend-George theory is fully developed, decaying turbulence which is indicated by a decreasing turbulence intensity. Therefore, the data points
in the near wake prior to the local maximum of the turbulence intensity will be masked in the following. In the plots, this is indicated by hatched areas.

### 3.2.2 Taylor Reynolds number

In order to apply the Townsend-George theory, we are looking for a region with fully developed turbulence where $Re_\lambda > 200$
holds, as stated in requirement 2a). In the appendix, the calculation of $Re_\lambda$ is detailed on equations A2, A3 and A4. In figure 3, $Re_\lambda$ is therefore plotted in the downstream region where the turbulence intensity decays. Therefore, this requirement of the Townsend-George theory is fulfilled.

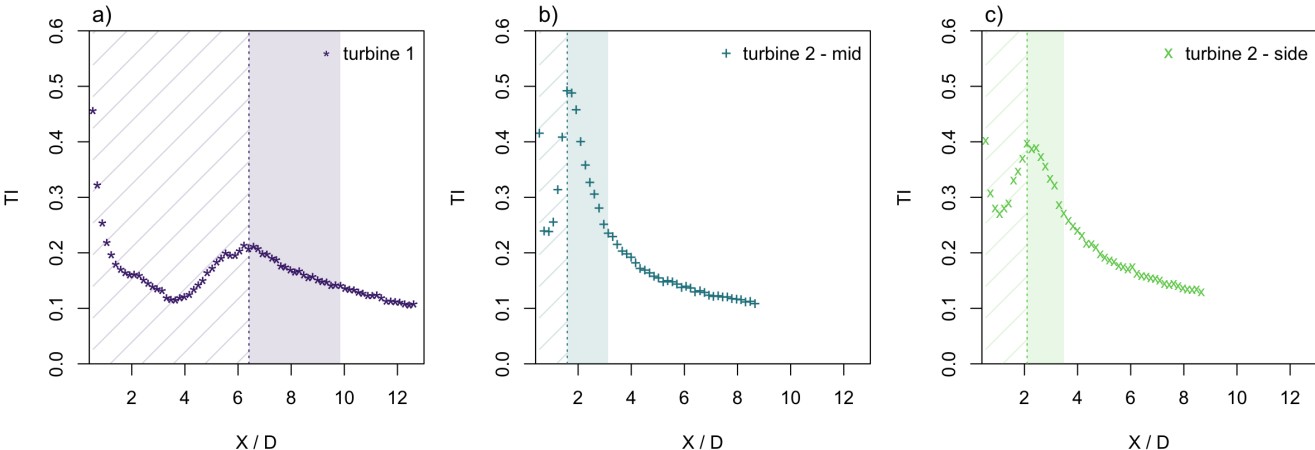

**Figure 2.** Centerline turbulence intensity $TI$ for the three different inflow conditions: a) turbine 1 - laminar inflow; b) turbine 2 mid - turbulent; c) turbine 2 side - turbulent and intermittent. The vertical lines mark the maximum of the $TI$ ($X/D = 6.24$ for turbine 1, $X/D = 1.79$ for turbine 2 mid, and $X/D = 2.45$ for turbine 2 side), the hatched regions upstream of the turbulence intensity maximum mark the regions that will be omitted in the following, and the colored area the region where the Taylor Reynolds number is changing (cf. figure 3).

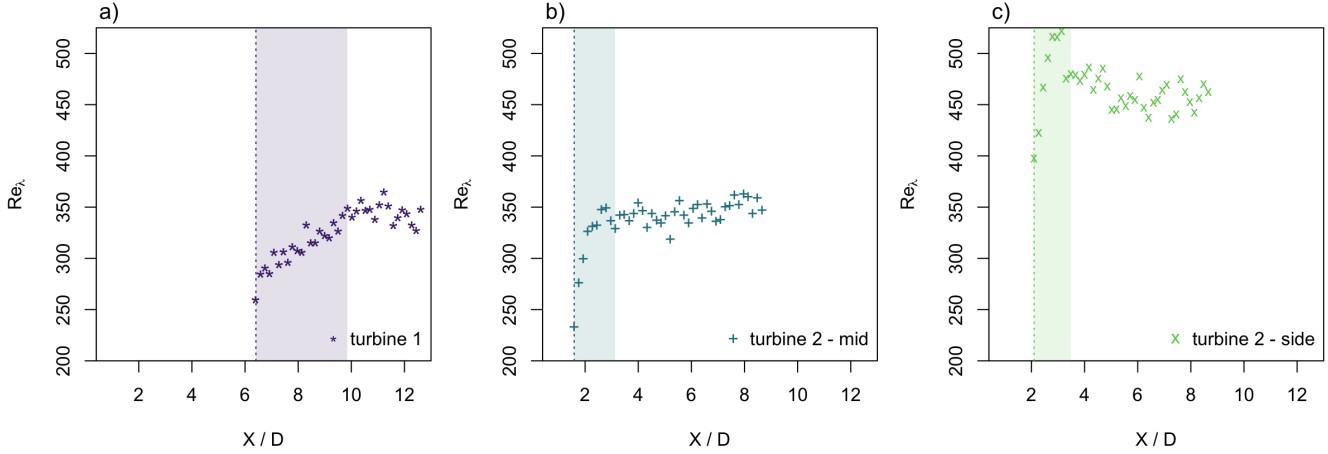

**Figure 3.** Downstream evolution of the Taylor Reynolds number $Re_\lambda$ for the three different inflow conditions at the centerline in the decay region of the $TI$: a) turbine 1 - laminar inflow; b) turbine 2 mid - turbulent; c) turbine 2 side - turbulent and intermittent. The vertical lines mark the maximum of the $TI$ and the colored area the region where the Taylor Reynolds number is changing.

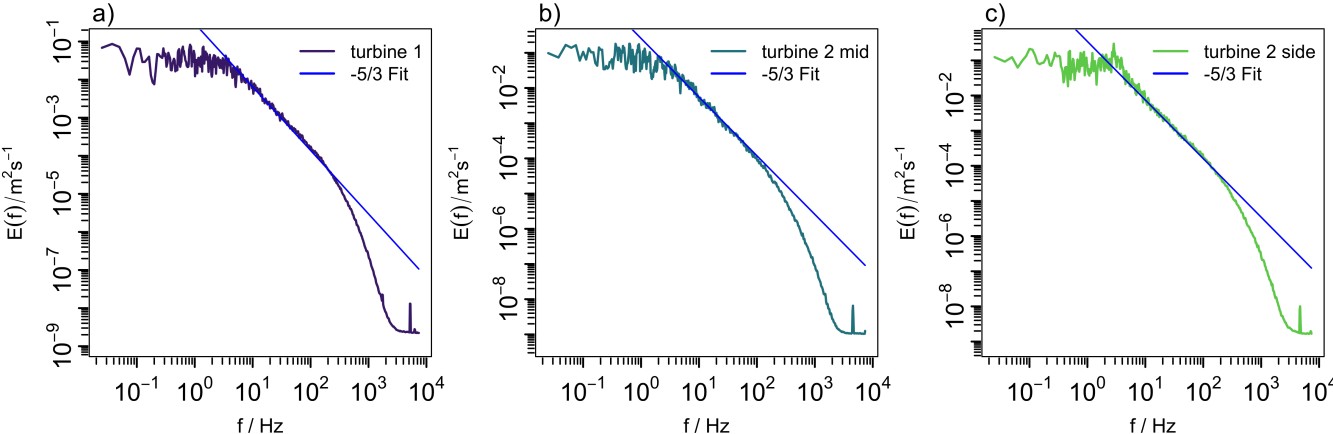

**Figure 4.** Energy spectral density $E(f)$ over frequency $f$ at the maximum turbulence intensity ($X/D = 6.24$ for turbine 1, $X/D = 1.79$ for turbine 2 mid, and $X/D = 2.45$ for turbine 2 side) for the three different inflow conditions: a) turbine 1 - laminar inflow; b) turbine 2 mid - turbulent; c) turbine 2 side - turbulent and intermittent. In blue, the $E(f) \propto f^{-5/3}$ decay is marked.

### 3.2.3 Energy spectral density

Next, the energy spectral density has to be checked. It is important to have an energy spectrum with a clear inertial sub-range that decays according to a power law, as explained in requirement 2b). In figure 4, the energy spectral density is plotted for the three inflow scenarios at the respective maximum of the turbulence intensity. Clearly, an inertial sub-range that decays according to $E(f) \propto f^{-5/3}$ is present in all three wake scenarios. In Neunaber et al. (2020), it is also shown how the energy spectral density has an inertial sub-range decaying with $E(f) \propto f^{-5/3}$ for all positions farther downstream. Therefore, the next criterion is fulfilled in the chosen downstream region.

### 3.2.4 Self-similarity

As the Townsend-George theory originates in classical bluff body wake analysis, another important assumption is that of the self-similarity of the wake, as stated in requirement 3. To verify the self-similarity in this study, the normalized velocity deficit $\frac{U_\infty - U(X,Y/\delta)}{U_\infty - U_c}$ is plotted over the normalized radial component $Y/\delta$. In the case of a self-similar behavior of the turbulence, the curves are expected to collapse. For the two cases turbine 1 and turbine 2 mid, cf. figure 5 a) and b), the curves collapse as required and thus, a self-similar behavior is confirmed. To show that the profiles collapse to a Gaussian curve as often implied when assuming a constant eddy viscosity, a fit according to

$$\frac{U_\infty - U(X,Y/\delta)}{U_\infty - U_c} = a \cdot \exp\left(-b \cdot (Y/\delta)^2\right) \tag{9}$$

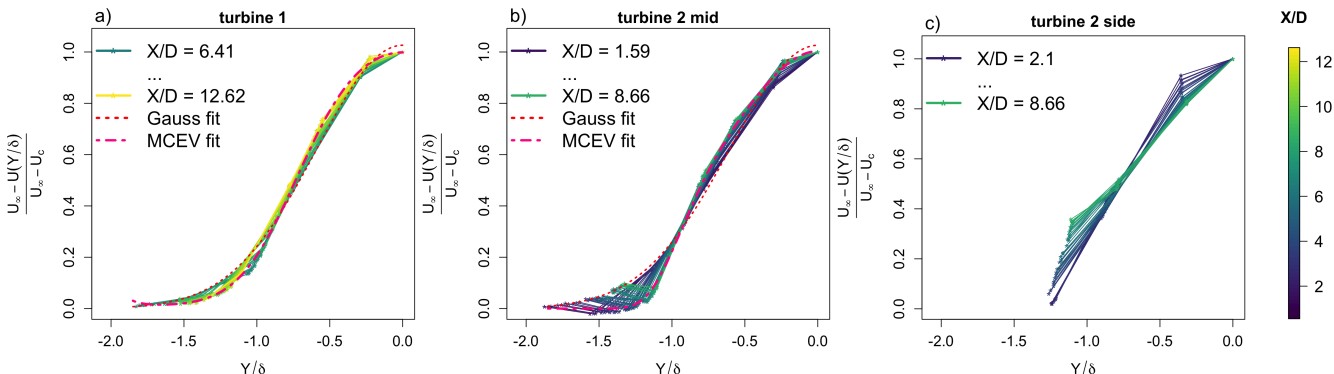

**Figure 5.** Plot of the wake deficit over the normalized span-wise position $Y/\delta$ for the three different inflow conditions: a) turbine 1 - laminar inflow; b) turbine 2 mid - turbulent inflow; c) turbine 2 side - turbulent and intermittent inflow. Only curves in the decay region of the turbulence intensity are included and the downstream position is indicated by means of the color map.

was applied. In addition, the modified constant eddy viscosity (MCEV) model discussed in Cafiero et al. (2020),

$$\frac{U_\infty - U(X, Y/\delta)}{U_\infty - U_c} = a \cdot \exp\left(-b \cdot (Y/\delta)^2 - c \cdot (Y/\delta)^4 - d \cdot (Y/\delta)^6\right) \tag{10}$$

has been applied. With root mean square (RMS) errors of 0.013 and 0.027 for turbine 1 and 2 mid, respectively, the MCEV model performs indeed better than the Gaussian fit with RMS errors of 0.033 and 0.054, respectively. In the asymmetric, more complex wake of turbine 2 side (cf. figure 5 c)), a self-similar behavior can not be found and therefore, we do not expect the Townsend-George theory to fully hold there.

As we present results obtained from 1d hot-wire anemometry, the test for self-similarity is restricted here to the mean velocity profile. However, Stein and Kaltenbach (2019) did investigate the self-similarity of the added Reynolds stress tensor components and the added turbulent kinetic energy in the wake of a model wind turbine exposed to an ABL profile. We assume therefore that this requirement also holds here.

### 3.2.5 Axisymmetry

In addition to self-similarity, also axisymmetry of the wake is required, as explained in requirement 4. As the measurements that we present have been carried out in one half of the wake, we are not able to directly verify the axisymmetry. However, based on the symmetric setups for turbine 1 and turbine 2 mid and other studies with similar conditions, see e.g. Stevens and Meneveau (2017), we conclude that the requirement of axisymmetry can be taken as valid for these inflow conditions.

It should be noted that the axisymmetry may be influenced by the presence of the ground and an ABL profile when investigating

the wake of a wind turbine in the field. However, as the mean far wake evolving downstream a turbine exposed to an ABL inflow is often described as the superposition of an ABL profile with an axisymmetric wake, it can be assumed that the requirement also holds for these cases (Bastankhah and Porté-Agel (2014); Stein and Kaltenbach (2019)).

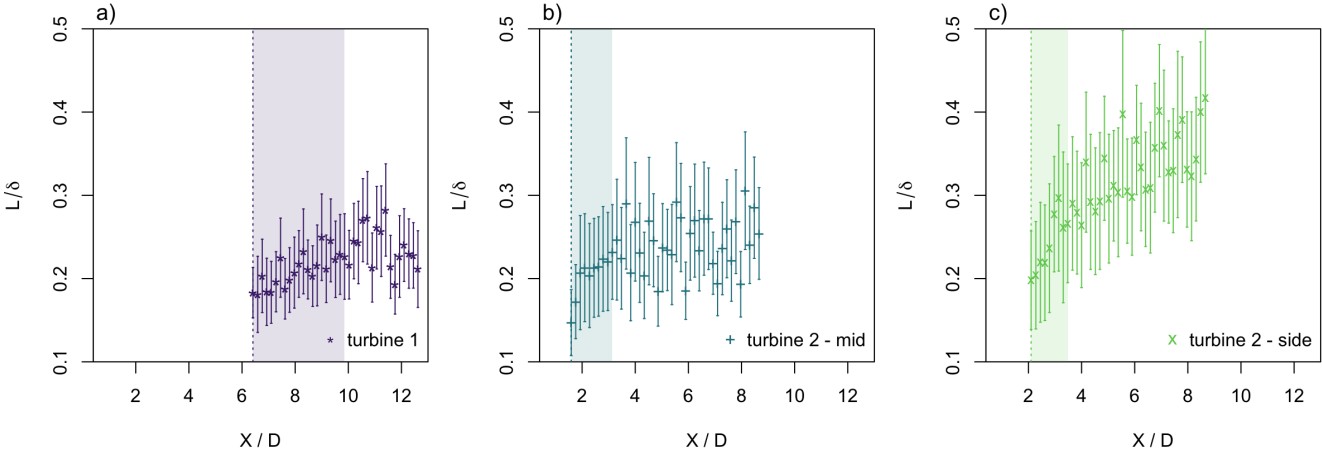

**Figure 6.** Downstream evolution of $L/\delta$ for the three different inflow conditions: a) turbine 1 - laminar inflow; b) turbine 2 mid - turbulent; c) turbine 2 side - turbulent and intermittent. The vertical lines mark the maximum of the $TI$ and the colored the area the region where the Taylor Reynolds number is changing.

### 3.2.6   Integral length scale over wake width

The last requirement that needs to be fulfilled is the behavior of $L/\delta$, see requirement 5. As explained above, $\delta \propto L$ is assumed in the Townsend-George theory for bluff body wakes. In figure 6, $L/\delta$ is therefore plotted over $X/D$ for the three scenarios, and error bars are included. The errors for $L/\delta$ were calculated using error propagation. As the calculation of L is quite sensitive, the error bars are rather large. Downstream of turbine 1 and turbine 2 mid, $L/\delta$ increases in the coloured region where $Re_\lambda$ changes, and it is approximately constant in the far wake. This shows how the wake turbulence has a tendency to develop towards symmetry. In the far wake, the data points are quite shattered because the integral length scale fluctuates in this region, especially downstream of turbine 2 where $L$ tends to fluctuate due to the turbulent background inflow (for the evolution of $L$, see Neunaber et al. (2020) and Neunaber (2019)). Downstream of turbine 2 side in the complex inflow, the points are scattered and an increase of $L/\delta$ is visible, but far downstream, $L/\delta$ appears to tend to a constant value. This indicates that despite the highly complex, sheared and asymmetric inflow the turbulence appears to drive the evolution of the wake towards symmetry. A deeper analysis of this will be postponed to future work. Briefly, we can conclude that the condition $L/\delta \approx const.$ is fulfilled in the far wake case of turbine 1 and turbine 2 mid but not in the case of turbine 2 side.

### Summary of chapter 3.2

In order to fulfill the first requirement of the Townsend-George theory, we first chose the downstream regions in which the turbulence intensity is decaying. In this region, the other requirements have been tested, and by means of the Taylor Reynolds number, the energy spectral density, the self-similarity and, particularly in the far wake, also the behavior of $L/\delta$, we can conclude that the Townsend-George theory can be applied to the wakes downstream of turbine 1 and turbine 2 mid. In the wake

**Table 1.** Summary of the investigation of which requirements are fulfilled at the centerline in the three wake scenarios: ✓ denotes the fulfillment of a criterion, (✓) the sufficient fulfillment of a criterion, and ✗ that the criterion is not fulfilled.

|      |                      | turbine 1 | turbine 2 mid | turbine 2 side |
| ---- | -------------------- | --------- | ------------- | -------------- |
| R1   | decaying turbulence  | $X/D \geq 6.24$ | $X/D \geq 1.79$ | $X/D \geq 2.45$ |
| R2 a) | $Re_\lambda > 200$  | ✓ | ✓ | ✓ |
| R2 b) | $E(f) \propto f^{-5/3}$ | ✓ | ✓ | ✓ |
| R3   | self-similarity      | ✓ | ✓ | (✓) |
| R4   | axisymmetry          | (✓) | (✓) | ✗ |
| R5   | $L \propto \delta$   | (✓) | (✓) | ✗ |

of turbine 2 side, the requirements are partially fulfilled (fully developed turbulence that is indicated by the decaying turbulence intensity, the high Taylor Reynolds numbers, and a decay of the energy spectral density has been verified while self-similarity, axisymmetry and $L/\delta \approx const.$ are not given). Table 1 summarizes this again. With the confirmation that the requirements are met for turbine 1 and 2 mid and partially for turbine 2 side, we will apply the Townsend-George theory to the data.

**4  Results**

After verifying the requirements that need to be fulfilled in order to apply the Townsend-George theory, the next step is to check whether non-equilibrium turbulence is present at the centerline. For this, we will first discuss the behavior of $Re_\lambda$ and $Re_L$ and afterwards, $C_\varepsilon$ will be investigated.

As stated above, the Taylor Reynolds number is supposed to change so that the presence of equilibrium and non-equilibrium
turbulence can be disentangled. In figure 3, it can be seen that $Re_\lambda$ changes in the region directly downstream of the local maximum turbulence intensity that was identified as the decay region in Neunaber et al. (2020) but remains then constant in the far wake. Therefore, non-equilibrium turbulence could be present in the region just after the turbulence intensity peak that is underlain in color ($6.24 \leq X/D \leq 9.86$ for turbine 1, $1.79 \leq X/D \leq 3.14$ for turbine 2 mid, and $2.45 \leq X/D \leq 3.48$ for turbine 2 side). We remark that $Re_\lambda$ is increasing in the region of interest, and while there is no specification in the Townsend-
George theory how $Re_\lambda$ has to change, George (1989) writes that $Re_L$ has to decrease. Therefore, we also show the downstream evolution of the local Reynolds number $Re_L$ in figure 7. It can be seen that $Re_L$ increases in the marked region where $Re_\lambda$ is also increasing. Farther downstream, $Re_L$ is constant downstream of turbine 1, and it decreases downstream of turbine 2 mid and side. Since $Re_\lambda$ remains constant in the far wake where $Re_L$ is decreasing, the results regarding the existence of non-equilibrium turbulence are inconclusive in that range. The results also suggest that we capture the onset of the streamwise
decay of $Re_L$ and $Re_\lambda$, in agreement with the theoretical prediction.

The next step is to additionally verify whether there is a dependence of $C_\varepsilon$ on the Taylor Reynolds number in order to differ between the equilibrium scaling and the non-equilibrium scaling. In figure 8, $C_\varepsilon(Re_\lambda)$ is therefore plotted for the three inflow conditions in the range where the turbulence intensity is decaying. $C_\varepsilon$ is calculated according to formula A5 in the appendix.

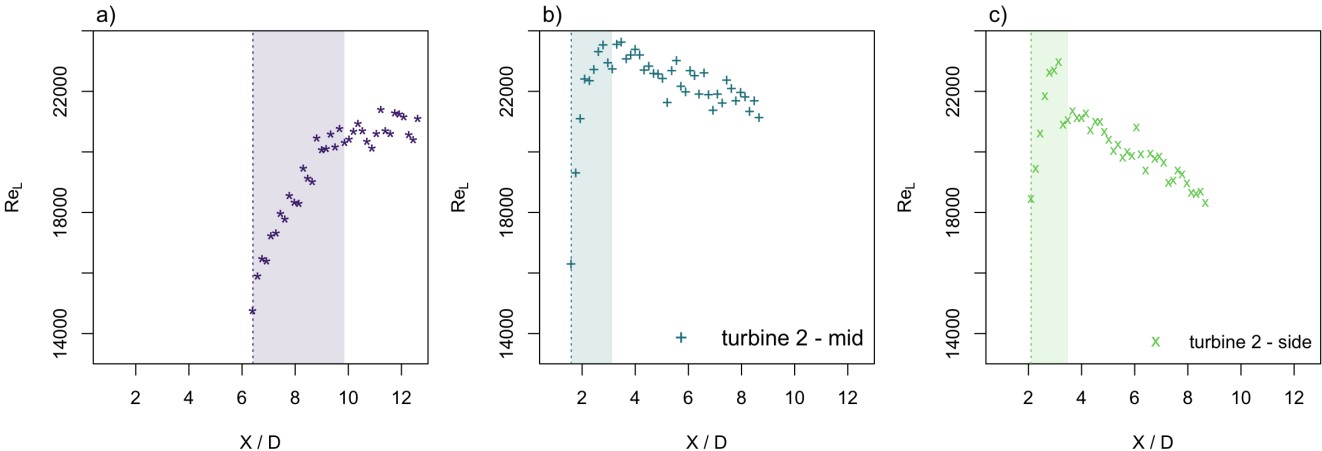

**Figure 7.** Downstream evolution of the local Reynolds number $Re_L$ for the three different inflow conditions at the centerline in the decay region of the $TI$: a) turbine 1 - laminar inflow; b) turbine 2 mid - turbulent; c) turbine 2 side - turbulent and intermittent. The vertical lines mark the maximum of the $TI$ and the colored area the region where the Taylor Reynolds number is changing.

The data points in the downstream region where $Re_\lambda$ is changing are plotted in color while the data points in the downstream region where $Re_\lambda$ is approximately constant are masked in grey, and the average over all masked points is plotted in blue. It can be seen that, while the values for $C_\varepsilon$ scatter about $\pm 15\%$ around the mean value, which is normal due to the sensitivity of the calculation of $C_\varepsilon$, no clear dependency of $C_\varepsilon(Re_\lambda)$ is present. Since this would be expected in the case of non-equilibrium turbulence, together with the inconclusive results from the investigation of $Re_\lambda$ and $Re_L$, we can conclude that equilibrium turbulence occurs and that we do not find evidence of non-equilibrium turbulence in any of the inflow scenarios in the evaluated region in the wake of a wind turbine.

It can also be seen that the three different wakes have a different average $C_\varepsilon$. This indicates inhomogeneity of the dissipation constant of the turbulence inside of a wind farm. Another consequence is that the dissipation may be instationary: When the wind direction changes, the wake of an upstream turbine may pass over a downstream turbine with the consequence that $C_\varepsilon$ in the inflow changes e.g. from the wake to the ABL inflow. In such a scenario $C_\varepsilon$ changes with time.

These consequences show the importance of the analysis of $C_\varepsilon$ in the wake of a wind turbine since computational fluid dynamics models normally rely on a constant dissipation coefficient.

After verifying all requirements and testing for the occurrence of non-equilibrium turbulence, finally, figure 9 shows the downstream evolution of the normalised velocity deficit $\Delta U/U_\infty$ over $X/D$ at the centerline for all inflow conditions. An error for the velocity deficit has been calculated from error propagation where the error of $U(X)$ comes from the calibration uncertainty and the error of $U_\infty$ is a statistical estimation of the inflow variation. The error bars are included in the plot but the error bars are very small, with the maximum relative errors being 0.0054 in the case of turbine 1, 0.0035 in the case of turbine 2 mid and 0.0043 in the case of turbine 2 side. In the following, we apply the EQ and NEQ scalings from the Townsend-George theory to the data without any preference. In addition, the two introduced wind turbine wake models, namely the

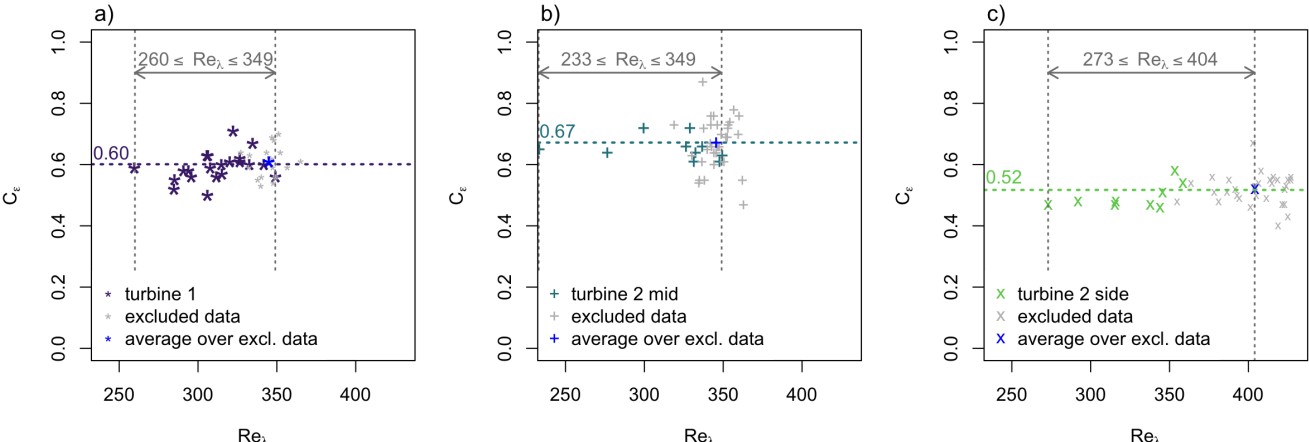

**Figure 8.** $C_\varepsilon(Re_\lambda)$ at the centerline for the three different inflow conditions: a) turbine 1 - laminar inflow; b) turbine 2 mid - turbulent; c) turbine 2 side - turbulent and intermittent. In grey, data points within the downstream region where $Re_\lambda$ does not change are masked and their mean value is plotted in blue.

Jensen model and the BP model, are fitted to the data. Here, we simplify the fitting of the EQ and NEQ scalings by using
$\Delta U = A_{EQ} \cdot (X - X_{0,EQ})^{-2/3}$ and $\Delta U = A_{NEQ} \cdot (X - X_{0,NEQ})^{-1}$ where $A$ and $X_0$ are fitted. For the Jensen and the BP models, the wake expansions $k_J$ and $k_{BP}$ are used as fit variables. The residual standard errors of the respective fits are given in the legends in figure 9. The fit parameters can be found in table 2. Note that we do not apply superposition wake models for the wakes of turbine 2 mid and side here but treat the wakes individually because we are interested in the difference a turbulent inflow has on the fit. With the hypothesis that a final universal turbulence state can be reached within a wind farm where multiple wakes are overlapping, the modeling of these multiple wake scenarios is not a question of superposition but rather of how and where this final turbulence state is reached. In this philosophy, the investigation of the individual wakes is thus of interest.

In the case of turbine 1, the NEQ wake model and the EQ wake model predict a similar evolution of the mean centerline velocity deficit and show the best results based on the residual standard error. As the maximum error for this data set is $\partial \Delta U / U_\infty = 0.0054$, we can conclude that they perform similarly. The BP wake model performs less well and it can be seen that there are deviations in the evolution of the normalized velocity deficit. The Jensen model does not capture the evolution of the mean centerline velocity deficit. It should be noted that, since the Jensen model uses a top-hat shaped wake to calculate the average wake velocity, it was never designed to accurately model the centerline velocity deficit. To give a specific example of the consequence of making an error in the estimation of the mean velocity, we will look at the BP model: In the beginning, the velocity deficit is underestimated by 10% and at the end of the measurement range, the velocity deficit is overestimated by 3%. This would lead to a difference in the power estimation of 25% respectively 10% based on the centerline velocity for this specific wake scenario.

Downstream of turbine 2 mid, it can be seen that the EQ model captures the evolution of the velocity deficit best. The NEQ

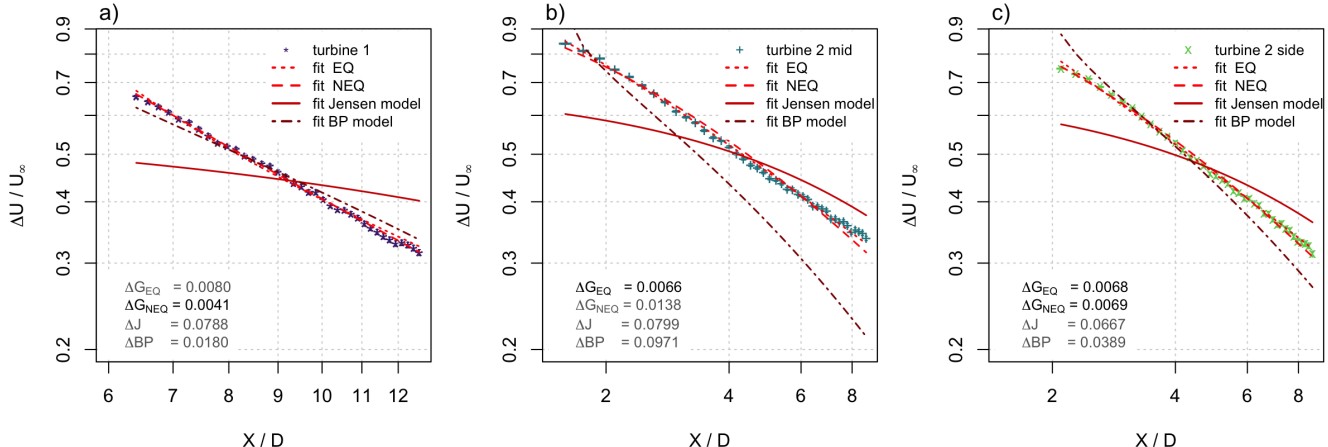

**Figure 9.** $\Delta U/U_\infty$ for three different inflow conditions: a) turbine 1 - laminar inflow; b) turbine 2 mid - turbulent; c) turbine 2 side - turbulent and intermittent. Error bars are included but very small. Different models are fitted to the recovery region, and the residual standard errors are presented. Note that the axes are logarithmic.

model performs less well and shows slight deviations from the data. In a systematic way, the model overestimates the velocity up to $X/D \approx 2.5$, underestimate the velocity in the range $2.5 \leq X/D \leq 6$ and overestimate it from $X/D \approx 6$. In the far wake, the velocity is overestimated by 3% which would result in an over-prediction of the power of approximately 9% for this specific wake scenario. The fits for the Jensen and the BP model show large deviations.

Downstream of turbine 2 side, the EQ and NEQ models perform best and fit the measurement points well. The Jensen model does not capture the evolution of the mean centerline velocity deficit. The BP model is not suitable for this wake condition with complex inflow, and the deviations are significant: at the end of the measurement range, the velocity is overestimated by approximately 6% which leads to an overestimation of the power by approximately 17%.

Overall, it is obvious from the presented results that the EQ wake model performs well for all three wake scenarios, and in combination with the result from above that $C_\varepsilon \approx const.$ (cf. figure 8), we conclude that the EQ model from classical wake theory is the wake that performs quite well. Note that the EQ model clearly improves the mentioned engineering models.

## 5   Discussion

In this work, we have introduced a different approach to model the wakes of wind turbines that originates in the theory of bluff body wakes. For this, we verified first that the requirements necessary to apply these theories are fulfilled (see requirements 1-5). Then, we searched for turbulence with non-equilibrium scaling by investigating the evolution of $C_\varepsilon$. While we did not find evidence for non-equilibrium turbulence, we could show that both the EQ and the NEQ bluff body wake theories fit the wind turbine wake remarkably well. While it is hard to disentangle the difference in the power-law exponents from the EQ and NEQ models (in the end, they are both similar in the limited streamwise range studied), we show that they both predict a

**Table 2.** Fit parameters and RMS errors for the different models predicting the evolution of $\Delta U/U_\infty$ for the three scenarios at the centerline.

|  |  | turbine 1 | turbine 2 mid | turbine 2 side |
|---|---|---|---|---|
| EQ | $A_{EQ}$ | 1.438 | 1.455 | 1.385 |
|  | $X_{0,EQ}/D$ | 3.290 | -0.654 | -0.300 |
|  | $\Delta G_{EQ}$ | 0.008 | 0.007 | 0.007 |
| NEQ | $A_{NEQ}$ | 3.751 | 3.614 | 3.437 |
|  | $X_{0,NEQ}/D$ | 0.751 | -2.798 | -2.457 |
|  | $\Delta G_{NEQ}$ | 0.004 | 0.014 | 0.007 |
| Jensen | $k_J$ | 0.008 | 0.020 | 0.022 |
|  | $\Delta J$ | 0.079 | 0.080 | 0.067 |
| BP | $k_{BP}$ | 0.013 | 0.030 | 0.024 |
|  | $\Delta BP$ | 0.018 | 0.097 | 0.039 |

functional form that properly fits our data: a power law with a no-zero virtual origin. Although the power law approach is not new to the description of the mean velocity deficit, see e.g. Porté-Agel et al. (2020), we add new physical aspects to the use as the formulas used here are derived from certain assumptions, and we test the requirements needed to apply the theory before
using it.

To further expand the discussion of the turbulence present in the wake of a wind turbine, we did calculate $C_\varepsilon$ for all measurement positions in the three wake scenarios in the regions where the energy spectral density shows an inertial sub-range with a power law decay. The results are presented as interpolated contour plots in figure 10 for the three wake scenarios. In addition, the positions of the blade tips are marked as horizontal red dashed lines, and the position of turbine 2 is marked as a vertical
red dashed line in the wake of turbine 1. For all wakes, $C_\varepsilon$ does not show strong variations inside the wake. At the centerline downstream of turbine 1 and turbine 2 mid, $C_\varepsilon$ is larger than around the blade tip position. Downstream of turbine 2 side, the variations of $C_\varepsilon$ inside the wake are even less pronounced and the value is smaller as above-mentioned. The variation of $C_\varepsilon$ due to the inflow turbulence means that different dissipation scalings are present inside a wind farm as already discussed in section 4, and during a change of the wind direction, even a temporal fluctuation of the dissipation scalings can occur. Another very
interesting effect is visible at some points in these plots: at the wake edges at the turbulent–non-turbulent interface between the wake and the laminar inflow, $C_\varepsilon$ is significantly higher than inside the wake. While the data presented here is not sufficient to draw conclusions, we suspect that this may indicate a ring of large $C_\varepsilon$ surrounding the wake, similar to the ring of high intermittency that was found to surround the wake in Schottler et al. (2018) and that was shown to be traceable along the whole measured range in Neunaber et al. (2020). However, further investigation on this topic is needed to confirm this. This shows
that beside the wake core that is characterized by equilibrium turbulence (indicated by $C_\varepsilon \approx const.$ around the centerline), there are distinctive regions with non-trivial turbulence that can be found in the wake of a wind turbine. One consequence of the variation of $C_\varepsilon$ in the wake is that the dissipation changes radially. Even with the complex inflow of turbine 2 side that is also exposed to this $C_\varepsilon$ ring, the wake center shows equilibrium turbulence. This confirms again the finding that the turbine

creates an own, dominant type of turbulence (Neunaber et al. (2020)).


With the knowledge that the wake core is characterized by equilibrium turbulence, we also applied the different wake models to the evolution of the velocity deficit at another radial position in the wake core, $Y/D = -0.21$. For the BP model, we only use the centerline velocity deficit part of equation 7. The results are presented in figure 11 for the three wake scenarios, and the coefficients are given in table 3. The plots include error bars that are obtained similarly to the errors in figure 9, and the
maximum errors are 0.0062 for turbine 1, 0.0077 for turbine 2 mid, 0.0066 for turbine 2 side and, in addition, 0.0055 for turbine 2 side at $Y/D = 0.31$. It can be seen that all wake models except the Jensen model perform very well in the case of turbine 1. With respect to the errors, the NEQ and BP models perform similarly and the EQ model shows a slightly worse fit. The Jensen model does not capture the evolution of the centerline velocity deficit, which is expected as it was developed to estimate wake losses for the whole wake. In the case of turbine 2 mid, the EQ model performs best, and the NEQ shows slight systematic
differences, as discussed above. Both the BP and the Jensen model do not capture the evolution centerline velocity deficit, but the Jensen model performs reasonably well in the far wake.

In the case of turbine 2 side, two radial positions, $Y/D = -0.21$ and $Y/D = 0.31$, have been used to test the wake models. It can be seen that in this complex flow, the models have greater difficulties to perform well, and the evolution of the mean velocity deficit is not captured very accurately in most cases. At $Y/D = -0.21$, the bluff body wake theories perform best.
For $Y/D = 0.31$, the error is the smallest for the EQ and NEQ models and the velocity is predicted well at the end of the measurement range. In the cases of the Jensen model and the BP model, it can be seen that the evolution of the velocity deficit is not captured very well which is also indicated by the RMS errors. Overall, this test illustrates that even though most of the models have been designed to fit the centerline velocity deficit, they also perform quite well in the wake core at inner radial positions. This is an interesting result because it is not easy to measure along the centerline in field measurements, and with
the concept of a wake core as introduced in Neunaber et al. (2020), the application of these models is expanded.

To further test the applicability of the models, in the next step, we will calculate the velocity deficit at the radial position $Y/D = -0.21$ (and additionally $Y/D = 0.31$ in the case of turbine 2 side) from a radial correction: The BP model already has an implemented radial correction. The Jensen model is a top-hat shaped model and we therefore forego a radial correction. The EQ and NEQ models can be expanded by including the radial profile information from equation 10 to calculate the normalized
velocity deficit $\Delta U(Y/D)/U_\infty$,

$$\frac{\Delta U(Y/D)}{U_\infty} = A_{EQ} U_\infty \left((x - x_{0,EQ})/\theta\right)^{-2/3} \cdot a \cdot \exp\left(-b(Y/\delta)^2 - c(Y/\delta)^4 - d(Y/\delta)^6\right) \tag{11}$$

and

$$\frac{\Delta U(Y/D)}{U_\infty} = A_{NEQ} U_\infty \left((x - x_{0,NEQ})/\theta\right)^{-1} \cdot a \cdot \exp\left(-b(Y/\delta)^2 - c(Y/\delta)^4 - d(Y/\delta)^6\right) \tag{12}$$

from the fit parameters obtained for the centerline velocity deficit and the wake profile. For this, the evolution of the wake
width has to be known, and it is given by equations 2 and 4. Here, we apply the equations to the evolution of $\delta(X/D)$, which is presented in figure 12 for the three scenarios, and we apply in addition a linear fit of the form $\delta(X/D) = \alpha + \beta \cdot (X/D)$.

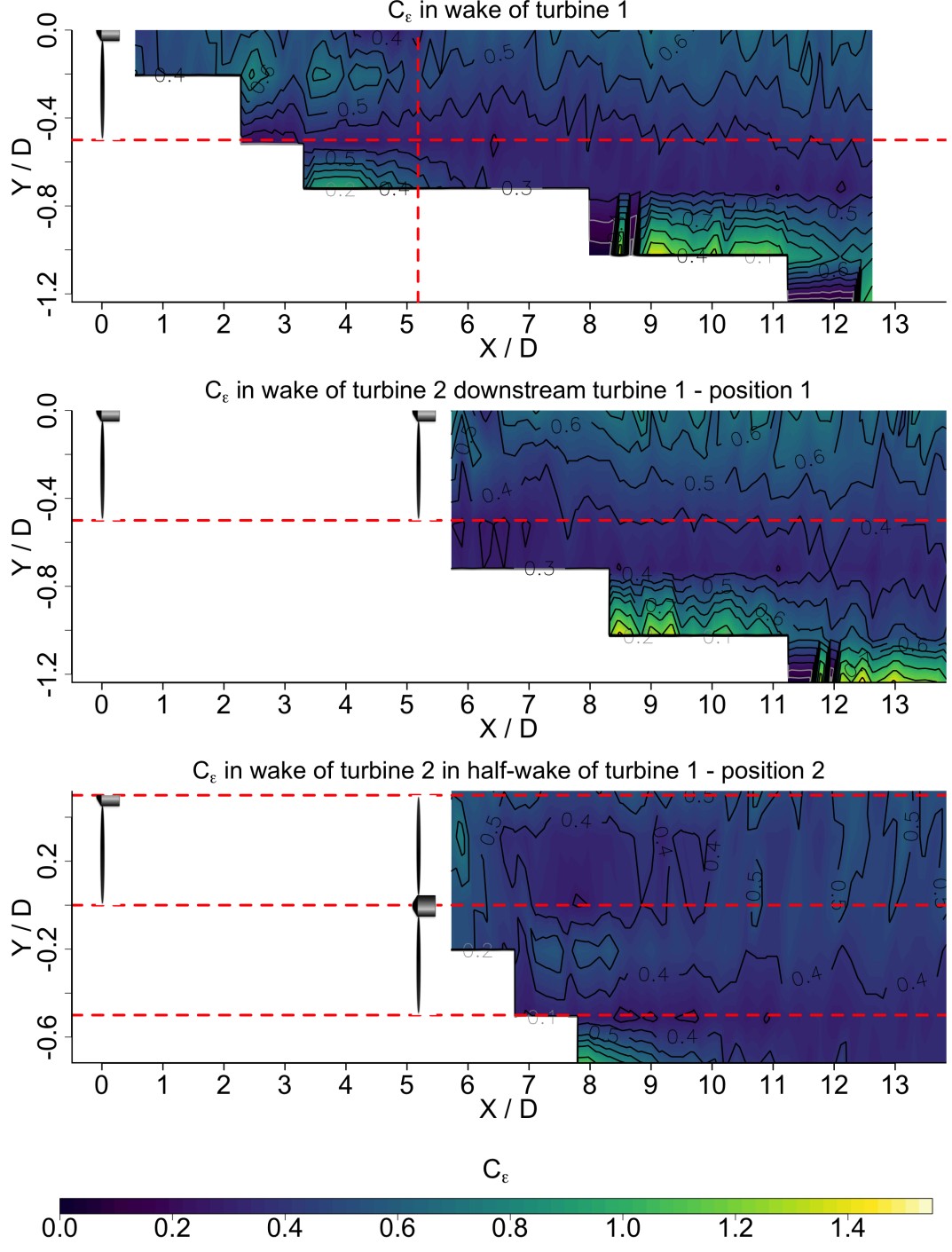

**Figure 10.** Interpolated surface plot of the downstream evolution of $C_\varepsilon$ for the three different inflow conditions: a) turbine 1 - laminar inflow; b) turbine 2 mid - turbulent; c) turbine 2 side - turbulent and intermittent. Outside of the turbulent wake, $C_\varepsilon$ is masked.

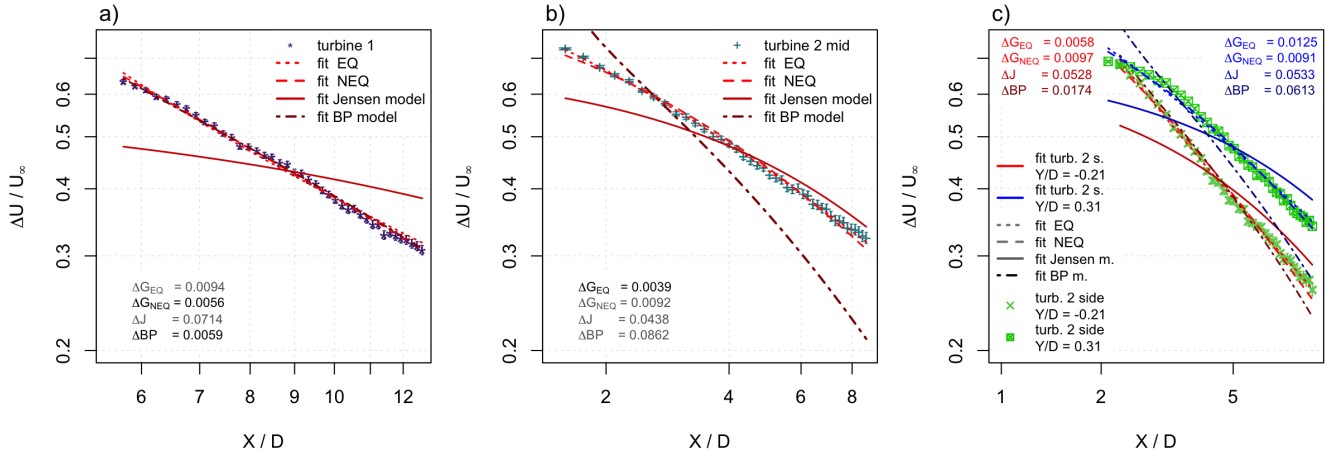

**Figure 11.** $\Delta U/U_\infty$ for the three different inflow conditions at the span-wise position $Y/D = -0.21$: a) turbine 1 - laminar inflow; b) turbine 2 mid - turbulent; c) turbine 2 side - turbulent and intermittent. Different models are fitted to the recovery region, and the residual standard errors are presented. Note that the axes are logarithmic.

**Table 3.** Fit parameters and RMS errors for the different models predicting the evolution of $\Delta U/U_\infty$ for the three scenarios at the radial position $Y/D = -0.21$. Note that for turbine 2 side, in addition, the parameters for the radial position $Y/D = 0.31$ are included.

|        |                | turbine 1 $Y = -0.21D$ | turbine 2 mid $Y = -0.21D$ | turbine 2 side $Y = -0.21D$ | turbine 2 side $Y = 0.31D$ |
|--------|----------------|------------------------|----------------------------|-----------------------------|----------------------------|
| EQ     | $A_{EQ}$       | 1.510                  | 1.474                      | 1.069                       | 1.587                      |
|        | $X_{0,EQ}/D$   | 2.237                  | -1.305                     | 0.364                       | -1.103                     |
|        | $\Delta G_{EQ}$ | 0.009                 | 0.004                      | 0.006                       | 0.013                      |
| NEQ    | $A_{NEQ}$      | 4.079                  | 3.870                      | 2.525                       | 4.177                      |
|        | $X_{0,NEQ}/D$  | -0.582                 | -3.870                     | -1.476                      | -3.707                     |
|        | $\Delta G_{NEQ}$ | 0.006               | 0.009                      | 0.010                       | 0.009                      |
| Jensen | $k_J$          | 0.009                  | 0.024                      | 0.031                       | 0.020                      |
|        | $\Delta J$     | 0.071                  | 0.044                      | 0.053                       | 0.053                      |
| BP     | $k_{BP}$       | 0.015                  | 0.030                      | 0.027                       | 0.023                      |
|        | $\Delta BP$    | 0.006                  | 0.086                      | 0.017                       | 0.061                      |

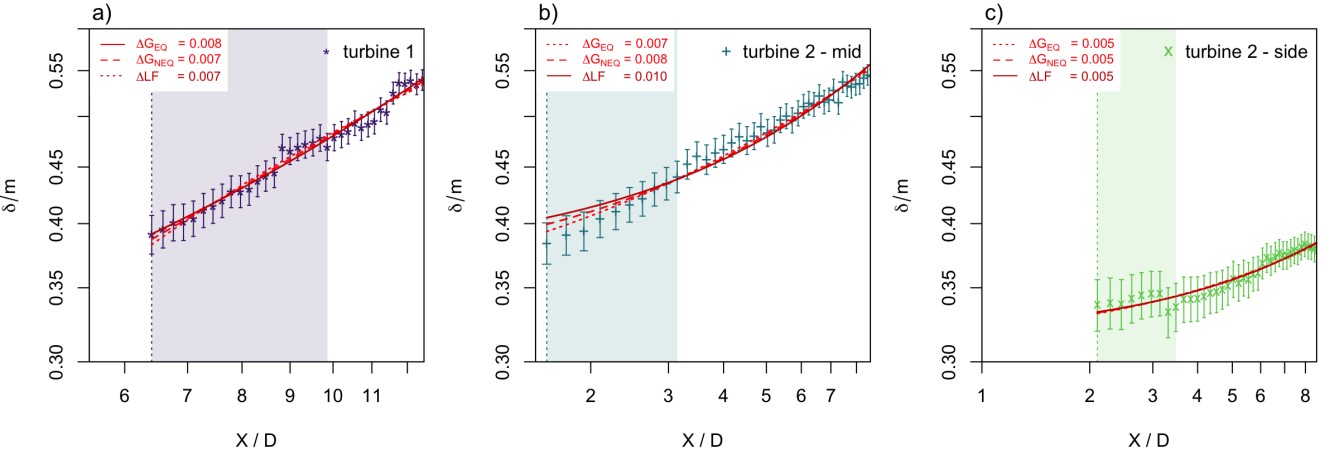

**Figure 12.** $\delta(X/D)$ for the three different inflow conditions: a) turbine 1 - laminar inflow; b) turbine 2 mid - turbulent; c) turbine 2 side - turbulent and intermittent. The EQ and NEQ models and a linear. Note that the axes are logarithmic.

It can be seen that all three models perform very similarly in the measurement range, and also, that $\delta(X/D)$ increases almost linearly. From the fit parameters given in table 4, we see that the fit coefficients $B_{EQ}$ and $B_{NEQ}$ of turbine 1 and turbine 2 mid are similar. As expected, the results are different for turbine 2 side where the prediction of the wake width is not very accurate

due to the complex inflow. In addition, it can be seen that the virtual origins are different from the ones obtained for the fits of the normalized velocity deficits. Therefore, while the Townsend George theory overall holds for a wind turbine wake, there are certain details that deviate and that will need further investigation in the future.

In figure 13, the normalized mean centerline velocity deficit at the radial position $Y/D = -0.21$ is plotted for the three scenarios, and the predicted evolutions are added for the different models. In the case of turbine 1, the EQ and NEQ models capture

the evolution very well from $X/D \approx 7$. The best fit is achieved with the NEQ model. The BP model captures the evolution of $\Delta U/U_\infty$ in the far wake but shows deviations closer to the rotor. As expected, the Jensen model does not perform well. In the case of turbine 2 mid, the EQ and NEQ models perform similarly well and capture the evolution of $\Delta U/U_\infty$ in the measurement range, but the curves start to deviate far downstream. The BP model shows a big deviation because already the centerline velocity deficit (cf. figure 9) was not captured well. The Jensen model also shows strong deviations. In the case of

turbine 2 side, $Y/D = -0.21$, the EQ and NEQ models do underpredict the velocity deficit in the far wake but the result is close to the measured data. The Jensen model is again strongly deviating. The BP model fits the evolution of $\Delta U/U_\infty$ best in this situation. In the case of $Y/D = 0.31$, none of the models perform well because all rely on a decreasing velocity deficit towards the free flow, but in the sheared inflow situation, the velocity deficit actually increases.

As we have seen, the EQ and NEQ bluff body wake models are significantly better in predicting the normalized mean ve-

locity deficit at $Y/D = -0.21$ than the engineering wind turbine wake models. One reason is that more parameters need to be obtained from fits of the wake shape and the centerline velocity deficit. Another reason is the implied virtual origin that is common in free shear flow analysis as it includes the transition of the wake to the final turbulent wake state. It gives these

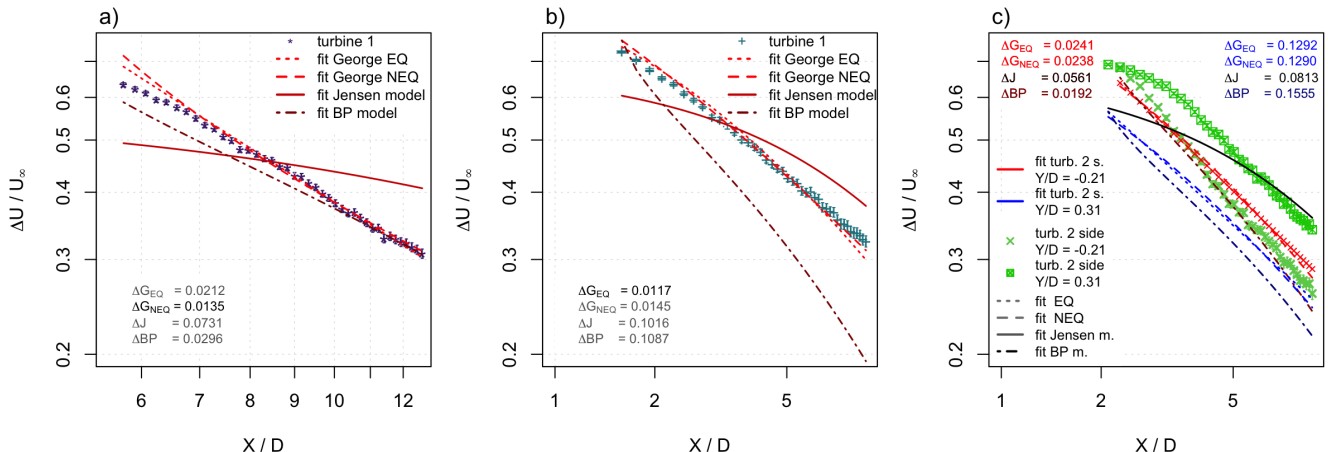

**Figure 13.** Modeled $\Delta U/U_\infty$ for the three different inflow conditions at the span-wise position $Y/D = -0.21$: a) turbine 1 - laminar inflow; b) turbine 2 mid - turbulent; c) turbine 2 side - turbulent and intermittent. Different models are fitted to the recovery region, and the residual standard errors are presented. Note that the axes are logarithmic.

**Table 4.** Fit parameters and RMS errors for the different models predicting the evolution of $\delta$ for the three scenarios.

|        |                 | turbine 1 | turbine 2 mid | turbine 2 side |
|--------|-----------------|-----------|---------------|----------------|
| EQ     | $B_{EQ}$        | 0.251     | 0.248         | 0.145          |
|        | $X_{0,EQ}/D$    | 2.875     | -2.401        | -0.873         |
|        | $\Delta G_{EQ}$ | 0.008     | 0.007         | 0.005          |
| NEQ    | $B_{NEQ}$       | 0.152     | 0.145         | 0.076          |
|        | $X_{0,NEQ}/D$   | -0.114    | -5.963        | -17.258        |
|        | $\Delta G_{NEQ}$| 0.007     | 0.008         | 0.005          |
| linear | $a$             | 0.468     | 0.482         | 0.359          |
|        | $b$             | 0.277     | 0.294         | 0.097          |
|        | $\Delta LF$     | 0.007     | 0.010         | 0.005          |

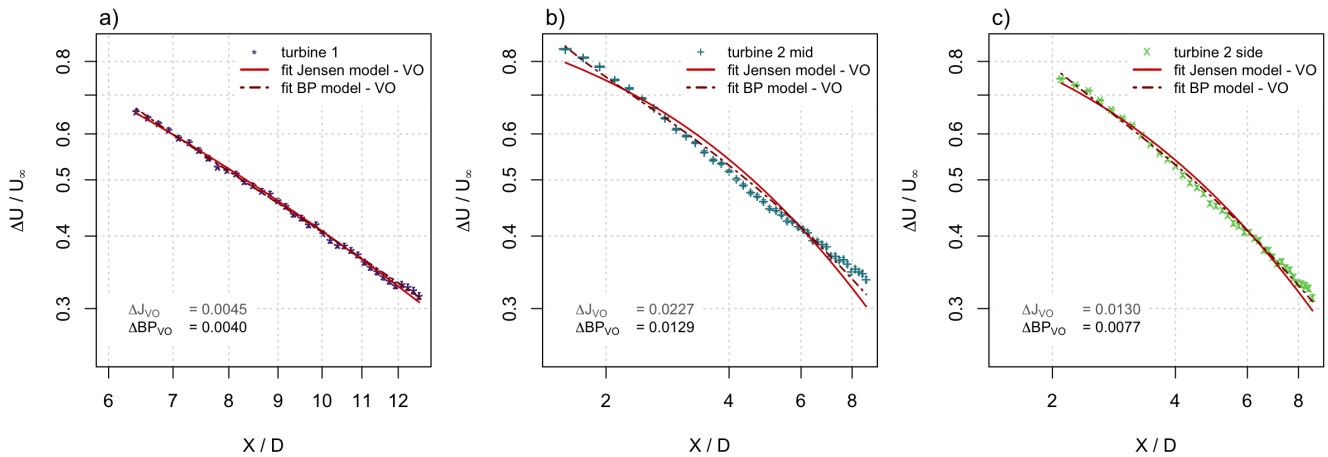

**Figure 14.** $\Delta U/U_\infty$ for the three different inflow conditions at the centerline: a) turbine 1 - laminar inflow; b) turbine 2 mid - turbulent; c) turbine 2 side - turbulent and intermittent. The Jensen and the BP models are modified by including a virtual origin in the fits. Note that the axes are logarithmic.

**Table 5.** Fit parameters and RMS errors for the altered Jensen and BP models with virtual origins predicting the evolution of $\Delta U/U_\infty$ for the three scenarios at the centerline.

|  |  | turbine 1 | turbine 2 mid | turbine 2 side |
|---|---|---|---|---|
| Jensen VO | $k_J$ | 0.035 | 0.041 | 0.042 |
|  | $X_{0,J}/D$ | 7.129 | 2.490 | 2.518 |
|  | $\Delta J$ | 0.005 | 0.023 | 0.013 |
| BP VO | $k_{BP}$ | 0.016 | 0.017 | 0.018 |
|  | $X_{0,BP}/D$ | 1.460 | -1.410 | -1.026 |
|  | $\Delta BP$ | 0.004 | 0.013 | 0.008 |

models a high flexibility to adapt to different wakes. While the concept of a virtual origin has been taken into account in the form of the initial wake width in other wake models, e.g. the Bastankhah-Porté-Agel model, a "true" virtual origin that is not

limited to a value of the order of magnitude of the turbine diameter improves the flexibility. Therefore, we checked whether the Jensen and BP models can be improved by including a virtual origin, thus substituting $X$ by $X - X_0$. The results are shown in figure 14, and the fit parameters are given in table 5. For both models, the implementation improves the accuracy of the fits significantly so that they are now in the order of magnitude of the errors obtained for the EQ and NEQ models in some cases. Nevertheless, a similar systematic deviation can be identified in the wakes of turbine 2 mid and side where the centerline mean

velocity deficit is overpredicted in one part of the wake and underpredicted in an other part of the wake. A clear difference in $X_0$ can be seen for laminar and turbulent inflow. This shows the potential of the concept of a virtual origin to include the background turbulence conditions.

## 6 Conclusions

In this paper we have introduced a new approach to look at the evolution of the wake of a wind turbine, the Townsend-George theory that has originally been derived from bluff body wakes to predict the evolution of the mean centerline velocity deficit in a laminar incoming flow. For this, we performed wind tunnel measurements of the wake downstream of a single turbine exposed to laminar inflow, and a turbine exposed to the wake of an upstream turbine and the half-wake of an upstream turbine. First, we have presented a systematic study on the applicability of the Townsend-George theory to wind-turbine-generated wakes for the different incoming flows. For this, we explored the streamwise range in which all requirements are fulfilled, cf. table 1. We found that all five requirements are (sufficiently) satisfied in the cases of turbine 1 and turbine 2 mid. The requirements are partially fulfilled in the case of turbine 2 side. Next, we studied the change of $Re_\lambda$ and $Re_L$ with downstream distance to interpret the behavior of $C_\varepsilon$ for this flow and to draw conclusions regarding the centerline evolution of $\Delta U$ and $\delta$ (i.e. equilibrium or non-equilibrium scaling). By means of $C_\varepsilon(Re_\lambda) \approx const.$, we found evidence for equilibrium turbulence in the investigated parts of the wake for all three scenarios. In the following step, we applied the Townsend-George theory of $\Delta U$ to estimate the velocity deficit in the three wake scenarios using the equilibrium and the non-equilibrium predictions. In addition, two engineering wind turbine wake models have been applied to the data. It was found that the Townsend-George theory does describe the wake of a wind turbine and even a second-row turbine very well. In particular, the equilibrium scaling gives good results for all three wake scenarios. Thus, the nature of the energy cascade is closer to the equilibrium scaling (i.e. the scalings consistent with the Richardson-Kolmogorov cascade) than to the non-equilibrium scaling for the different wind turbine wake flows. Especially in the more complex situation of a wake inflow, the equilibrium bluff body wake model outperforms the wind turbine wake models despite the fact that the former has been derived for the wake evolving downstream of a static bluff body exposed to uniform laminar inflow. This opens up a new perspective as the wake of a wind turbine does behave in principal similar in different inflow conditions. While multiple wake situations are normally solved by using wake superposition models (which are not in accordance with the Navier Stokes equations), with our interpretation, a final turbulence state within a wind farm due to the turbulence evolution processes is plausible. A more precise description of wakes within a wind farm is of big interest as due to the cubic dependence of the power on the wind velocity, even rather small deviations in the velocity estimation can lead to large deviations in the power prediction.

To further investigate the applicability of the wake models, we also applied them to a different radial position, $Y/D = -0.21$. We saw that the equilibrium bluff body wake model again outperforms the common wind turbine wake models. As the centerline is not always captured perfectly in field measurements, this result is very interesting for the application. Also, we could show that a radial correction can be used to predict the wake at $Y/D = -0.21$ using the fit coefficients obtained for the centerline. In comparison with the BP model that has a built-in radial correction, the Townsend-George theory shows again better results.

The virtual origin native to the Townsend-George theory was identified as one main advantage of the bluff body wake models. As mentioned above, this virtual origin differs from the concept of the initial wake width used in some of the engineering wake models in that it accounts for the turbulence transition (and therefore is common to several turbulence one-point quantity

scalings). Therefore, we also discuss how the implementation of a virtual origin may improve the Jensen and the BP models, especially as we see that it can be used to include the inflow turbulence in the wake models.

A final result of this analysis is that $C_\varepsilon$ does change in the wake both with respect to the radial position and with respect to the inflow condition. A consequence is that the turbulence dissipation inside a wind farm is inhomogeneous and instationary which needs to be considered in simulations to further improve wind farm modeling. The investigation of $C_\varepsilon$ can also be used as a tool to add information on the type of turbulence present, and, thus, has the potential to refine the understanding of turbulence within a wind farm.

Overall, this work shows in a detailed manner the consistent applicability of the equilibrium turbulence bluff body wake model from turbulence theory to wind turbine wakes, and thus the importance of correctly taking into account $C_\varepsilon$. While the Jensen model is derived from conservation of mass, and the BP model includes conservation of mass and momentum and considers the shape of the wake, we show that the additional inclusion of the turbulent nature is crucial. Nevertheless, we also see that the models fail in multi-wake scenarios where the inflow is not symmetric and where thus superposition approaches would as well fail. This and the additional dependence of the wind turbine wake on the operational state of the turbine are difficulties occurring inside find farms that need further investigation.

For the future, a more detailed investigation of $C_\varepsilon$ inside a wind farm will be interesting in order to identify possible regions with non-equilibrium turbulence and their dependence on the inflow turbulence and the operational state of the turbine (e.g. the thrust coefficient). In addition, the possibility of expanding engineering wind turbine wake models with a virtual origin should be further explored, especially with respect to the inflow turbulence. With the extremely asymmetric example of turbine 2 side, we could show that the application of the equilibrium wake model works to a certain degree because the turbulence tends to evolve towards a symmetric wake as far as possible. This gives a good indication for new theories to model half-wake and multi-wake scenarios with asymmetric inflows where $C_\varepsilon$ can be included.

Another open challenge will be to combine the ideas discussed here with the broader topic of wind farm wakes which are for example discussed in Platis et al. (2020) and Volker et al. (2015) who successfully "use the classical wake theory (Tennekes and Lumley (1972)) to describe the sub-grid-scale wake expansion".

*Acknowledgements.* The authors would like to thank the German Environmental Foundation, DBU, for funding the project with a PhD scholarship (nr. 20014/342) for Ingrid Neunaber. A special thanks goes to the research group Turbulence, Wind Energy and Stochastic at the University of Oldenburg for helpful discussions and experimental cooperation, in particular Michael Hölling, Jannik Schottler who built the model wind turbines, and Vlaho Petrović who designed the control.

*Author contributions.* IN acquired the data, performed the initial analysis and data investigation. IN and MO performed further data analysis and wrote the original draft. IN and JP acquired the funding. IN, JP and MO developed the methodology, and reviewed and edited the manuscript.

*Competing interests.* The authors declare that they have no conflict of interest.

*Data availability.* The data are available upon request.

## Appendix A: Estimation of turbulence quantities

To estimate the integral length scale $L$ and the Taylor length $\lambda$, we use the one-dimensional energy spectra as proposed by Hinze (1975). As the upper boundary of the inertial sub-range. the integral length scale can be estimated calculating the limit of the energy spectrum in the frequency domain for $f$ approaching 0,

$$L = \lim_{f \to 0} \left( \frac{E(f) \cdot U}{4u'^2} \right). \tag{A1}$$

$u'$ is the RMS value of the fluctuating streamwise velocity $U(t)$.

To estimate an error for the integral length scale, we use the standard deviation $\sigma_E$ of the points of the spectrum that are used to estimate the limit $\lim_{f \to 0}$ which is the "flat" part of the spectrum at low frequencies. $\sigma_E$ is then expressed in % of the mean energy in these frequencies, $\sigma_{E,n} = \sigma_E / \langle E \rangle$, and $\Delta L = L \cdot \sigma_{E,n}$.

The Taylor length is defined as

$$\lambda = \left( \frac{u'^2}{\langle (\partial U'/\partial X)^2 \rangle} \right)^{1/2}, \tag{A2}$$

where $U' = U(t) - U$ are the fluctuations of the velocity. $\langle \cdot \rangle$ denotes the ensemble average. The energy spectrum in the space domain can be used to calculate $\langle (\partial U'/\partial X)^2 \rangle$,

$$\left\langle \left( \frac{\partial U'(X)}{\partial X} \right)^2 \right\rangle = \int_{k_{min}}^{k_{max}} k^2 E(k) \mathrm{d}k, \tag{A3}$$

with the wave number $k$.

By means of the Taylor length $\lambda$, the RMS value of the fluctuating streamwise velocity $u'$ and the kinematic viscosity $\nu$, the Taylor Reynolds number is calculated,

$$Re_\lambda = \frac{u' \cdot \lambda}{\nu}. \tag{A4}$$

The energy dissipation rate $\varepsilon$ can be estimated by $\varepsilon = 15\nu \left\langle (\partial U'/\partial X)^2 \right\rangle$ (cf. Pope (2000)) under the assumptions of isotropy in small scales and the validity of Taylor's hypothesis of frozen turbulence. The derivative is calculated using formula A3.

Using the RMS value of the fluctuating streamwise velocity as estimate of the turbulent kinetic energy, $C_\varepsilon$ is calculated:

$$C_\varepsilon = \frac{\varepsilon L}{u'^3} = \frac{15L}{Re_\lambda \lambda}. \tag{A5}$$

The integral wake width is calculated in the following way,

$$\delta^2(X) = \frac{1}{\Delta U} \int_0^\infty (U_\infty - U)\, r \mathrm{d}r,$$ (A6)

and as the number of radial positions, six, is limited in this study, we calculate $\delta$ applying linear interpolation between the points. We do not use equation 9 or 10 as we wanted to avoid any bias that may occur due to fits deviating from the measured data. To estimate the error of $\delta$, $\Delta\delta$, we use the difference between $\delta$ calculated with linear interpolation and $\delta$ calculated from the six positions without any interpolation. This serves as an estimate of the maximum error.

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
