# Peer review of "Application of the Townsend-George theory for free shear flows to single and double wind turbine wakes - a wind tunnel study"

_Wind Energy Science, 2021_

## Referee Comment (RC1)

**Review of the manuscript wes-2021-13, entitled "Investigation of the dissipation in the wake of wind turbine array", by I. Neunaber, J. Peinke, M. Obligado.**

This manuscript leverages single-component hot-wire measurements collected through wind tunnel tests of downscaled wind turbine models to explore the potential of modeling the mean velocity deficit in wind turbine wakes and wake width through the classical theory of wakes for bluff bodies developed by Townsend and George. In the Introduction, this theory is qualitatively described, while the empirical model of the wake velocity deficit and wake width as a function of the downstream location is provided for both equilibrium and non-equilibrium turbulence in Sect. 2.1. The wake models used as a benchmark, namely the Jensen and the Gaussian wake models are introduced in Sect. 2.2. The experimental setup is reported in Sect. 3.1. A key part of the manuscript is Sect. 3.2, where the authors attempt to verify the requirements for the Townsend-George theory. Eventually, these requirements seem to be fulfilled in the far-wake of the case turbine 1, turbine 2, and only partially for turbine 2 -side. However, the proposed model will be applied to all three cases.

The results seem to indicate that the proposed model fits well with the experimental data, with the equilibrium case having a smaller error than the non-equilibrium case. The authors suggest considering a virtual origin for the application of wake models, which is not a novel feature for wind turbine wake models.

My main comments are:
- The clarity and sharpness of the statements and discussion should be largely improved throughout the manuscript. This is particularly important in Sect. 3.2 when assessing the requirements of the Townsend-George theory. The authors should report graphically or in a table for what range of the wake and flow case each requirement is fulfilled. It should be clarified when a complete equilibrium is achieved and when non-equilibrium turbulence is considered. The discussion about the wake turbulence properties is interesting, yet it seems elusive rather than conclusive. For instance, I was not able to find details why the core of the wake is considered in equilibrium turbulence state.
- The validity of the power law to characterize downstream evolution of the wake velocity deficit is not new (see e.g. the review by F. Porte-Agel et al., BLM, 2020 and references therein). I am not sure if this work actually provides more predictable capabilities, than what is known about the use of power laws for prediction of wake features.
- The use of a virtual origin is not a new feature for a wake model, see e.g. Ishihara T, Qian G. A new Gaussian-based analytical wake model for wind turbines considering ambient turbulence intensities and thrust coefficient effects. J. Wind Eng. Ind. Aerodyn. 2018;177:275-292, and reference therein. Connections to previous works should be provided about this wake feature.

More comments are reported below, which I hope might help to revise the current manuscript.

**Comments:**

1.      L1-6, These statements seem more suitable for an introduction rather than an abstract. I would suggest sharpening the focus of the abstract highlighting the research strategy and the results obtained.

2.      L16-17, you can clarify that wake interactions may occur only under certain wind conditions, namely wind direction, incoming wind speed and stability regimes. Wake interactions do not occur in a continuous fashion. Please add some references on this topic.

3.      L35, Please clarify the following statement. What do you mean for "*…the shear layers surrounding the wake have me*t"?

4.      L106, cross-check the equation for the momentum thickness. I believe the first term in the integral should be $U$ rather than $U_\infty$.

5.      L110-120, can you provide concisely the difference in flow properties for an equilibrium and non-equilibrium turbulence?

6.      L221, "*In addition, the Taylor Reynolds number is supposed to change so that the presence of equilibrium and non-equilibrium turbulence can be disentangled.*" This statement is not really clear. Do you mean that status of non-equilibrium turbulence occurs when the Taylor Reynolds number varies with *x*? Please clarify.

7.      Sect. 3.2.2 should be re-written and clarified. Express clearly the conditions for $Re_\lambda$ and $Re_L$ and state for each flow case under what wake regions those are satisfied.

8.      Fig. 5. Please specify the locations where these velocity signals were collected.

9.      L 265, "*The errors for L/δ were calculated using error propagation.*" Please provide more details on the mentioned error propagation method.

10.     At the end of section 3, maybe where now there is a not-numbered section denote "Summary" I would add a table or a sketch summarizing the results of this analysis for each flow case, namely the five requirements and the wake regions where those are fulfilled.

11.     Figs. 9, 11 and related text, I am not sure applications of the Jensen and BP models have been done properly. Typically, wake velocity fields are calculated for individual wind turbines; then, an overlapping wake model is used to predict the wake flow in presence of wake interactions. Is this the procedure you applied? Please provide details.

12.     L 353, I am not sure you discussed the detection of a turbulent/non-turbulent interface for the wake. Please clarify, in case I missed this discussion.

13.     L359-360, what flow properties did you leverage to infer that the wake center shows equilibrium turbulence.

14.     Fig. 10 and related text (L355 - 360). This mentioned ring with enhanced values of $C_\varepsilon$ is a bit elusive for all the three flow cases investigated. The region with higher $C_\varepsilon$ is located at the boundary of the measurement domain, which makes it difficult to assess if this is actually a ring or an artifact due to the interaction between the wake and the background wind tunnel flow. Any comment clarifying this comment would be beneficial.

15.     L 426-426, please specify for what wake region the five requirements of the Townsend-George theory are fulfilled for each flow case.

16.     L 429-430, specify what feature of $C_\varepsilon$ provides evidence for equilibrium evidence, in what wake region and for what flow case.

17.     L454, please revise the text reporting that the Jensen model was formulated only considering mass conservation (JensenNO.Anoteonwindgeneratorinteraction.Risø-M1983)

18. L446, The use of a virtual origin for a wind turbine wake model is not a novel concept, see e.g. Ishihara T, Qian G. A new Gaussian-based analytical wake model for wind turbines considering ambient turbulence intensities and thrust coefficient effects. J. Wind Eng. Ind. Aerodyn. 2018;177:275-292, and reference therein.

---

## Author Comment (AC1)

We would like to thank Stefan Emeis for taking the time to read and react to our manuscript. In the following, we would like to address some of his criticisms.

The manuscript by Neunaber et al. investigates the wakes of one and two turbines in a wind tunnel study. Unfortunately, the title and the abstract tell a competely different story. The wind tunnel is not mentioned at all in the abstract. In this title and abstract are completely misleading.

This has been addressed and the title and the abstract have been adapted accordingly.

This is a pity, because the idea conveyed in the abstract is interesting as well. There exists a large body of literature on turbulent wakes behind flow obstacles since decades. To use part of this information for today's wind turbine wake research is desirable and also challenging. But, unfortunately, this does not seem to be the major topic of this manuscript. It only presents a comparison of one of these older wake theories with a wind tunnel study.

We agree that the abstract may have given the wrong impression that there were many wake theories, and we have changed it accordingly. We would like to point out that we are successfully applying the only analytical solution that exists today for the self-preserving axisymmetric turbulent wake to the wake of a wind turbine. While the theory itself is not new, this study gives valuable insight into how the description of a wake from a turbulence point of view can help to improve the understanding of the turbulence within a wind turbine wake and adapt engineering wake models.

Sentence number 4 of the abstract reads " However, although wind turbine wakes have been subject to various studies, they are still not fully understood." But no references are given. There have been large field experiments in recent years in order to learn about wakes behind larger offshore wind turbine arrays (the title of this manuscript says that this is a study on wakes of wind turbine arrays!). E.g., Platis et al. (2000) give an overview on what was achieved in the offshore wind farm wake experiment WIPAFF in the North Sea where aircraft conducted in situ measurements within the farm wakes. A general overview on onshore and offshore wind turbine wake experiments could be obtained from Sun et al. (2020)

- o In this manuscript, we are investigating the wakes of a single turbine and a wind turbine exposed to the wake of an upstream turbine, which we now clearly point out in the title and in the abstract. We agree that we were not sufficiently specific in the first version of the paper. As these wake scenarios occur inside a wind farm, they should not be mixed with the investigation of whole wind farm wakes. However, it would definitely be interesting to see in the future whether parts of Townsend-George theory could also be used to increase the understanding of the wake of a wind farm. We therefore picked up this point in the conclusion of our manuscript.
- o As this manuscript focuses on the application of the Townsend-George theory on wind turbine wakes by using wind tunnel data, we do not include a general review of wind turbine wake experiments or wake models, as this has been done by other works.

Also not mentioned are modelling studies, e.g., those by Fitch et al. (2012) or Volker et al. (2015). The experiments mentioned before and the model simulations fit together in many aspects. I.e., quite a lot has been learned about wind turbine wakes in recent years.

While we agree that a lot of interesting results have been published in the past years, as indicated above, the focus of this work is on what can be learned from the application of the Townsend-George theory to wind turbine wakes. Also, as already mentioned, we do not include a general

review of wind turbine wakes but include specific information where needed and give references. As we agree that the description of the wind farm wake is an interesting point, it is now commented on in the conclusions.

As already stated, we agree that the focus of our work on the wake of a single turbine for different inflow conditions was not clearly stated in the previous version of the manuscript. Definitely, for the wide context of wakes in farms, these are excellent papers.

I therefore would like to suggest a major revision of this manuscript. It could turn out to become a highly interesting paper in the end covering a highly up-to-date subject in renewable energy research.

References

Fitch, A. C., Olson, J. B., Lundquist, J. K., Dudhia, J., Gupta, A. K., Michalakes, J., & Barstad, I. (2012). Local and mesoscale impacts of wind farms as parameterized in a mesoscale NWP model. Mon. Wea. Rev., 140, 3017-3038.

Platis, A., Bange, J., Bärfuss, K., Cañadillas, B., Hundhausen, M., Djath, B., ... & Emeis, S. (2020). Long-range modifications of the wind field by offshore wind parks–results of the project WIPAFF. Meteorol. Z., 29, 355-376. DOI: 10.1127/metz/2020/1023
This citation has been included in the manuscript.

Sun, H., Gao, X., & Yang, H. (2020). A review of full-scale wind-field measurements of the wind-turbine wake effect and a measurement of the wake-interaction effect. Renew. Sustain.Energy Rev.,132, 110042. DOI: 10.1016/j.rser.2020.110042
This citation has been included in the manuscript

Volker, P. J. H., Badger, J., Hahmann, A. N., & Ott, S. (2015). The Explicit Wake Parametrisation V1. 0: a wind farm parametrisation in the mesoscale model WRF. Geosci. Model Devel., 8, 3715-3731.
This citation has been included in the manuscript

---

## Author Comment (AC2)

**Reviewer 1**
**Review of the manuscript wes-2021-13, entitled "Investigation of the dissipation in the wake of wind turbine array", by I. Neunaber, J. Peinke, M. Obligado.**

This manuscript leverages single-component hot-wire measurements collected through wind tunnel tests of downscaled wind turbine models to explore the potential of modeling the mean velocity deficit in wind turbine wakes and wake width through the classical theory of wakes for bluff bodies developed by Townsend and George. In the Introduction, this theory is qualitatively described, while the empirical model of the wake velocity deficit and wake width as a function of the downstream location is provided for both equilibrium and non-equilibrium turbulence in Sect. 2.1. The wake models used as a benchmark, namely the Jensen and the Gaussian wake models are introduced in Sect. 2.2. The experimental setup is reported in Sect. 3.1. A key part of the manuscript is Sect. 3.2, where the authors attempt to verify the requirements for the Townsend-George theory. Eventually, these requirements seem to be fulfilled in the far-wake of the case turbine 1, turbine 2, and only partially for turbine 2 -side. However, the proposed model will be applied to all three cases. The results seem to indicate that the proposed model fits well with the experimental data, with the equilibrium case having a smaller error than the non-equilibrium case. The authors suggest considering a virtual origin for the application of wake models, which is not a novel feature for wind turbine wake models.

We would like to thank the reviewer for taking the time to review our paper. His/her feedback does help to improve the quality of the paper by adding to its clarity and readability. In the following, we detail how we addressed each specific comment.

*My main comments are*:

• The clarity and sharpness of the statements and discussion should be largely improved throughout the manuscript. This is particularly important in Sect. 3.2 when assessing the requirements of the Townsend-George theory. The authors should report graphically or in a table for what range of the wake and flow case each requirement is fulfilled. It should be clarified when a complete equilibrium is achieved and when non-equilibrium turbulence is considered. The discussion about the wake turbulence properties is interesting, yet it seems elusive rather than conclusive. For instance, I was not able to find details why the core of the wake is considered in equilibrium turbulence state.

We took this comment into account and re-evaluated the manuscript accordingly. We agree that the addition of a table to summarize the fulfillment of each requirement helps the readability and added it on page 14. Also, we removed any discussion about possible occurrence of non-equilibrium turbulence from chapter 3.2 so that only the general requirements are discussed.
Also, we improved the clarity of the identification of equilibrium and non-equilibrium turbulence by adding the criteria that need to be checked in chapter 2.1 in a similar manner as the requirements, cf. page 6. In accordance with these criteria, the results are checked for the occurrence of non-equilibrium turbulence at the beginning of chapter 4, where we first investigate the behavior of the Taylor Reynolds number and the local Reynolds number finding inconclusive results (ll. 323) and then look at the behavior of $C\varepsilon$ finding evidence for equilibrium turbulence (ll. 332, l. 483).
We also state more precisely in ll. 404,
"*This shows that beside the wake core that is characterized by equilibrium turbulence (indicated by $C\varepsilon \approx const.$ around the centerline), there are distinctive regions with non-trivial turbulence that can be found in the wake of a wind turbine.*"

We hope that this additionally helps to clarify the statements that are made regarding the occurrence of equilibrium and non-equilibrium turbulence.

• The validity of the power law to characterize downstream evolution of the wake velocity deficit is not new (see e.g. the review by F. Porte-Agel et al., BLM, 2020 and references therein). I am not sure if this work actually provides more predictable capabilities, than what is known about the use of power laws for prediction of wake features.

We agree with the reviewer that the application of a power law to wind turbine wakes is not new and we point this out in the Discussion (ll. 387):
"*Although the power law approach is not new to the description of the mean velocity deficit, see e.g. Porté-Agel et al. (2020), we add new physical aspects to the use as the formulas used here are derived from certain assumptions, and we test the requirements needed to apply the theory before using it.*"
The novelty of our work therefore comes from the physical model that supports such power laws.

• The use of a virtual origin is not a new feature for a wake model, see e.g. Ishihara T, Qian G. A new Gaussian-based analytical wake model for wind turbines considering ambient turbulence intensities and thrust coefficient effects. J. Wind Eng. Ind. Aerodyn. 2018;177:275-292, and reference therein. Connections to previous works should be provided about this wake feature.

We agree that some wake models, as the Bastankhah-Porté-Agel model and the comparable model by Ishihara T and Qian G have a term like $k \cdot X/D + b$ where $k$ is defined as the wake growth rate and $b$ is an initial wake width that could also be interpreted as a virtual origin. $b$ should therefore be in the order of magnitude of the turbine diameter.
In contrast, in the Townsend-George theory, a *true* virtual origin that is depending on the flow's streamwise development is present. In it, the transition of the wake to its final turbulent state is taken into account, and it is not restricted. We clarify this in ll. 461 in the Discussion,
"*Another reason is the implied virtual origin that is common in free shear flow analysis as it includes the transition of the wake to the final turbulent wake state. It gives these models a high flexibility to adapt to different wakes. While the concept of a virtual origin does occur in the form of the initial wake width in other wake models, e.g. the Bastankhah-Porté-Agel model, a "true" virtual origin that is not limited to a value of the order of magnitude of the turbine diameter improves the flexibility.*"
And in ll. 504 in the Conclusion
"*The virtual origin native to the Townsend-George theory was identified as one main advantage of the bluff body wake models. As mentioned above, this virtual origin differs from the concept of the initial wake width used in some of the engineering wake models in that it accounts for the turbulence transition (and therefore is common to several turbulence one-point quantity scalings).*"

*More comments are reported below, which I hope might help to revise the current manuscript. Comments:*

1. L1-6, These statements seem more suitable for an introduction rather than an abstract. I would suggest sharpening the focus of the abstract highlighting the research strategy and the results obtained.

We rewrote the abstract. It now focuses on the research strategy and the results.

2. L16-17, you can clarify that wake interactions may occur only under certain wind conditions, namely wind direction, incoming wind speed and stability regimes. Wake interactions do not occur in a continuous fashion. Please add some references on this topic.

We clarified that wake interactions do not continuously occur and added a reference (ll. 19) "*Wind turbines are usually clustered in wind farms with the consequence that downstream turbines operate depending on the wind direction and the wind speed in the turbulent wakes of upstream turbines (e.g. Barthelmie et al. (2007); Sun et al. (2020)).* "

3. L35, Please clarify the following statement. What do you mean for "…the shear layers surrounding the wake have met"?

We agree the reviewer that the sentence was unclear. We did change the sentence (ll. 39) to: "*The far wake is typically identified as the part of the wake where the shear layers that evolve between the faster ambient flow and the lee of the object of investigation have met and the turbulence is fully developed.*"

4. L106, cross-check the equation for the momentum thickness. I believe the first term in the integral should be U rather than U_infty.

Thank you, we changed this.

5. L110-120, can you provide concisely the difference in flow properties for an equilibrium and non-equilibrium turbulence?

While we left this part unchanged, we listed the criteria for equilibrium and non-equilibrium turbulence on page 6 to clarify the difference:
„**Equilibrium/Non-Equilibrium turbulence Criteria.**
i.    *Does Cε=const. hold? In this case, no Reynolds number dependence should be seen.*
      yes*: equilibrium turbulence*
      no*: indication for non-equilibrium turbulence*
ii.   *The Taylor Reynolds number Reλ and the local Reynolds number ReL need to change with downstream distance in order to verify criterion i. More specifically, ReL has to decrease according to George (1989) in the case of non-equilibrium turbulence.*
      *If ReL and Reλ do not change, it is therefore not possible to draw conclusions on the occurrence of equilibrium and non-equilibrium turbulence and the results are inconclusive.*"

6. L221, "In addition, the Taylor Reynolds number is supposed to change so that the presence of equilibrium and non-equilibrium turbulence can be disentangled." This statement is not really clear. Do you mean that status of non-equilibrium turbulence occurs when the Taylor Reynolds number varies with x? Please clarify.

To clarify this part, we introduced criteria to distinguish between equilibrium and non-equilibrium turbulence on page 6.
Also, we now separate the discussion of the fulfillment of the requirements in chapter 3.2 and the investigation of the flow regarding equilibrium/non-equilibrium turbulence in the beginning of chapter 4 to improve the readability by not mixing the two parts of the investigation.

7. Sect. 3.2.2 should be re-written and clarified. Express clearly the conditions for $Re_\lambda$ and $Re_L$ and state for each flow case under what wake regions those are satisfied.

We re-wrote section 3.2.2. so that it contains now solely the discussion of the requirement $Re_\lambda$ > 200, and we shifted the discussion of $Re_\lambda(X/D)$ and $Re_L(X/D)$ to the beginning of chapter 4, the Results. There, we added also the wake regions

8. Fig. 5. Please specify the locations where these velocity signals were collected.

We added the locations in the caption of figure 4.

9. L 265, "The errors for $L/\delta$ were calculated using error propagation." Please provide more details on the mentioned error propagation method.

We now provide more information on the calculation of the individual errors of L and delta in the appendix.

10. At the end of section 3, maybe where now there is a not-numbered section denote "Summary" I would add a table or a sketch summarizing the results of this analysis for each flow case, namely the five requirements and the wake regions where those are fulfilled.

We added a table (table 1) on page 14 to summarize the results of the investigation of the requirements.

11. Figs. 9, 11 and related text, I am not sure applications of the Jensen and BP models have been done properly. Typically, wake velocity fields are calculated for individual wind turbines; then, an overlapping wake model is used to predict the wake flow in presence of wake interactions. Is this the procedure you applied? Please provide details.

The reviewer raises a fair point. Indeed, contrary to the often-used overlapping wake models, here we treat each wake as individual wake. We clarify this now in the new manuscript, that now reads (ll. 352):
"*Note that we do not apply superposition wake models for the wakes of turbine 2 mid and side here but treat the wakes individually because we are interested in the difference a turbulent inflow has on the fit. With the hypothesis that a final universal turbulence state can be reached within a wind farm where multiple wakes are overlapping, the modeling of these multiple wake scenarios is not a question of superposition but rather of how and where this final turbulence state is reached. In this philosophy, the investigation of the individual wakes is thus of interest.*"

12. L 353, I am not sure you discussed the detection of a turbulent/non-turbulent interface for the wake. Please clarify, in case I missed this discussion.

We did not, and clarified this in the text, it now reads (ll. 399):
"*Another very interesting effect is visible at some points in these plots: at the wake edges at the turbulent–non-turbulent interface between the wake and the laminar inflow, $C\varepsilon$ is significantly higher than inside the wake.*"

13. L359-360, what flow properties did you leverage to infer that the wake center shows equilibrium turbulence.

In the wake center, *Cε~const.* which, together with a change of the local Reynolds number *ReL* is a fulfilment of the criteria for equilibrium turbulence as stated in the added criteria on page 6. In addition, we specify now in ll. 404

*"This shows that beside the wake core that is characterized by equilibrium turbulence (indicated by Cε≈const. around the centerline), there are distinctive regions with non-trivial turbulence that can be found in the wake of a wind turbine."*

14. Fig. 10 and related text (L355 - 360). This mentioned ring with enhanced values of $C_$ is a bit elusive for all the three flow cases investigated. The region with higher $C_$ is located at the boundary of the measurement domain, which makes it difficult to assess if this is actually a ring or an artifact due to the interaction between the wake and the background wind tunnel flow. Any comment clarifying this comment would be beneficial.

We agree that our conclusion was drawn from experience rather from data evidence, and we therefore changed the text to clarify this. It now reads (ll. 399):
*"Another very interesting effect is visible at some points in these plots: at the wake edges at the turbulent–non-turbulent interface between the wake and the laminar inflow, Cε is significantly higher than inside the wake. While the data presented here is not sufficient to draw conclusions, we suspect that this may indicate a ring of large Cε surrounding the wake, similar to the ring of high intermittency that was found to surround the wake in Schottler et al. (2018) and that was shown to be traceable along the whole measured range in Neunaber et al. (2020). However, further investigation on this topic is needed to confirm this."*

15. L 426-426, please specify for what wake region the five requirements of the Townsend-George theory are fulfilled for each flow case.

We specify this now in table 1 on page 14.

16. L 429-430, specify what feature of $C_$ provides evidence for equilibrium evidence, in what wake region and for what flow case.

We clarify now in ll. 483
*"By means of Cε(Reλ)≈const., we found evidence for equilibrium turbulence in the investigated parts of the wake for all three scenarios"*
Which clarifies, as we hope, the passage together with the criteria from page 6.

17. L454, please revise the text reporting that the Jensen model was formulated only considering mass conservation (JensenNO.A note on wind generator interaction. Risø-M1983)

Thank you, we corrected this (l. 516)

18. L446, The use of a virtual origin for a wind turbine wake model is not a novel concept, see e.g. Ishihara T, Qian G. A new Gaussian-based analytical wake model for wind turbines considering ambient turbulence intensities and thrust coefficient effects. J. Wind Eng. Ind. Aerodyn. 2018;177:275-292, and reference therein.

As discussed in detail above, we agree on this and added in ll. 504 in the Conclusion
*"The virtual origin native to the Townsend-George theory was identified as one main advantage of the bluff body wake models. As mentioned above, this virtual origin differs from the concept of the initial wake width used in some of the engineering wake models in that it*

*accounts for the turbulence transition (and therefore is common to several turbulence one-point quantity scalings).''*

---

## Author Comment (AC3)

**Reviewer 2**

The paper investigates the applicability of equilibrium and non-equilibrium turbulence models from classical wake theories for wind turbine wake flows. Single hot-wire anemometry is used to characterise turbine wakes in three different scenarios: (i) a turbine subject to laminar inflow, (ii) one subject to full wake conditions, and finally (iii) a case in which the turbine is partially located in the wake of another turbine. The focus of the paper is given to verification of requirements for the validity of Townsend-George theory. The paper also studies in detail the evolution of wake centre velocity with streamwise distance and compare results of classical wake theories with the experimental data and common engineering wake models used in the wind energy community. Indeed, the subject of this work is interesting and relevant to the wind energy community. I appreciate the efforts of the authors to bridge the gap between turbulence research on wake flows in fluid mechanic community and the research on turbine wakes in the wind energy community. There are however major issues with suitability of experimental data and presentation of results. I will elaborate my comments in the following in the hope that it helps authors improve the quality of their manuscript:

We would like to thank the reviewer for evaluating our paper and for giving feedback that helped to improve the quality and readability of the manuscript. In the following, we will address every comment given below:

- Unsuitability of experimental dataset: I think the streamwise measurement range is too short which makes it very difficult to distinguish which model works better. For instance, in figure 12, all different relationships (x, x^1/3, x^1/2) seem to capture the variation of wake width with streamwise distance. At least, the authors could use log plots instead of linear plots for both velocity deficit and wake width plots. That way it would be much easier to find more systematically the exponent of a power function, for instance, whether the wake centre velocity deficit decreases with x^-2/3 based on Eq. 1 or with x^-1 based on Eq. 3.

  o While we agree that for a study of a classical bluff body wake, the range studied here would not be sufficient, we would like to point out that the range relevant for most wind energy applications is covered as the spacing of wind turbines rarely exceeds 8D.
  o We agree with the use of a logarithmic scale to improve the visualization of the deviation of different fits. Therefore, figures 9, 11,12,13 and 14 were changed and have now logarithmic axes.

- Comparison of model predictions: I think the way that the comparison with previous models has been made is not fair. First, EQ or NEQ models have two coefficients that can be tuned, whereas the other two models only have one empirical input. More importantly, it is not clear over which range of streamwise distance, fitting has been done. It is problematic if the whole range of [2D, 8D] is used to fit the model. The purpose of existing turbine wake models is to predict the far wake region, and these models are not expected to work well in the near wake region. For instance, in figure 9b, if you try to fit the BP and Jensen model only for [4D, 8D], their predictions should be improved.

- While we see the point of the reviewer that having two fit parameters that can be adapted is in advantage, we disagree with the statement that a comparison was "not fair": All engineering wake models are designed to describe the evolution of the wake of a wind turbine with increasing distance using only parameters that can be estimated from available information such as the current thrust coefficient and the inflow velocity. Therefore, while they have fewer free fit parameters, the used parameters are adapted to the flow and the turbine operation. They have been validated and calibrated to fit the wake of a wind turbine.

    In contrast, the EQ and NEQ model have been derived analytically from the perspective of flow physics and no calibration is applied to include the inflow or the turbine operation indirectly. Indeed, they are supposed to be valid for any generator, as far as an axisymmetric turbulent wake is produced.

- We agree that the models are *all* derived for the far wake – both the EQ and the NEQ wake models and the engineering wake models. We are aware that directly after the peak of the turbulence intensity, the turbulence may not be fully developed yet (for that reason, Neunaber (2020) uses the classification "decay region" for the transition to the far wake). However, from a practical standpoint, the description of as much of the wake as possible is of interest, especially the region starting from 2D downstream (in the case of turbulent inflow). In a wind farm, the spacing is often narrow (~3D-6D) and the ability of a wake model to also capture the wake of an upstream turbine in this situation is thus important. We therefore apply *all* wake models for the whole region in which the turbulence intensity decreases and the turbulence is already sufficiently evolved and compare the results in a similar manner. Our results, such as better performing power-law fits and the constancy of $C\varepsilon$, tend to point out that using far-wake mathematical tools in our flow remains approximately valid. There remain indeed some open questions and limitations that should be addressed in future works.

- Misleading title: there are two key words in the title: "dissipation" and "wind turbine array". None of them are really the main focus of this work. While C_epsilon is discussed in the paper, there is minimal discussion on dissipation in the turbine wake. Moreover, there is no more than two turbines used in this work. I therefore think it is a bit of stretch to use "wind turbine array" in the title.

    - We agree with the comment and changed the title to

        "*Application of the Townsend-George theory for free shear flows to single and double wind turbine wakes - a wind tunnel study*"

- Abstract should be more specific. The first half is more like an introduction talking about the importance of turbine wake studies. Also experimental setup and the data used to study these different models are not discussed in the abstract.

    - We thank the reviewer for this remark. We re-wrote the abstract following his/her recommendation. The quite long introduction part was removed in favor of a description of the methodology and the main results.

- Line 80: By placing a turbine in the wake of another turbine, the behaviour of a turbine within a turbulent background is studied. However, the turbulence generated by an

upwind turbine consisting of wake rotation and shedding vorticity is not expected to be identical to the one generated in the atmospheric boundary layer flow. Please clarify this either in line 80 or somewhere else in the manuscript.

- o We added a comment at ll. 94:
  *"As this work serves as a proof of the applicability of the Townsend-George theory, we do not include an investigation of the influence of an atmospheric boundary layer (ABL) profile where inflow characteristics may differ. However, it is generally assumed that the wake of a wind turbine in an ABL can be seen as a superposition between the ABL profile and the mean velocity deficit (cf. Bastankhah and Porté-Agel (2014)). This is in agreement with Neunaber et al. (2021) where the Townsend-George theory also gives good results in the case of field measurements obtained in the wake of a full-scale turbine using a LiDAR."*

- Section 3.2.2: For completeness, it would be useful to define the Taylor Reynolds number here.

  - o We decided not to define the Taylor Reynolds number here but to refer to the appendix where the definition is given and all relevant calculations are detailed on. The relevant formulas are now linked in 3.2.2.

- Figure 6: Self-similarity of velocity deficit profiles is examined here, but the self-similarity of shear-stress profiles should be also checked. I am conscious that with single hotwire anemometry, it is not possible to look into this. Ideally repeat some of your measurements with x-wire, or at least mention this as a limitation of the experimental setup.

  - o This is a valid point, and while a repetition of the measurements using an x-wire is out of the scope of this study, we now discuss this limitation of the setup in 3.2.4 but also reference the paper of Stein and Kaltenbach (2019) who investigate the self-similarity of added Reynolds stress tensor components and the added turbulent kinetic energy of a wake evolving downstream of a model turbine exposed to an ABL inflow. The passage now reads (ll. 275):
    *"As we present results obtained from 1d hot-wire anemometry, the test for self-similarity is restricted here to the mean velocity profile. However, Stein and Kaltenbach (2019) did investigate the self-similarity of the added Reynolds stress tensor components and the added turbulent kinetic energy in the wake of a model wind turbine exposed to an ABL profile. We assume therefore that this requirement also holds here."*

- Integral length scale: Final results seem to be quite sensitive to the value of the integral length scale. Did you try estimating its value via other methods, eg autocorrelation function? It is of interest to add a brief discussion on the impact of integral length scale evaluation on final results.

[Figure]

o  As shown in the figure above, we did calculate the integral length scale both using the autocorrelation (criterion: integration up to the first zero 0 crossing) and the energy spectrum. The results are similar so that we can conclude that a dependence of the results on the way of calculating L is not given.

o  In the paper, we do not explicitly discuss the sensitivity of the results to the value of the integral length scale as it is solely used in figure 6 for the purpose of checking the requirement $L\sim\delta$ (note that the local Reynolds number ReL is defined in the Introduction using the wake width and not the integral length scale). In figure 6, we include the error bars for this reason, and details on the error estimation for L and $\delta$ are now given in the appendix.

- Line 261: wake axisymmetry: Please add a brief discussion on how the presence of ground and boundary layer may affect the axisymmetry of the turbine wake in real situations.

o  We separated the sub-section on the axisymmetry in the revised manuscript and also added a discussion of the influence of the ground and an ABL, it now reads (ll. 279):
*"3.2.5  Axisymmetry*
*In addition to self-similarity, also axisymmetry of the wake is required, as explained in requirement 4. As the measurements*
*that we present have been carried out in one half of the wake, we are not able to directly verify the axisymmetry. However, based on the symmetric setups for turbine 1 and turbine 2 mid and other studies with similar conditions, see e.g. Stevens and Meneveau (2017), we conclude that the requirement of axisymmetry can be taken as valid for these inflow conditions. It should be noted that the axisymmetry may be influenced by the presence of the ground and an ABL profile when investigating the wake of a wind turbine in the field. However, as the mean far wake evolving downstream a turbine exposed to an ABL inflowis often described as the superposition of an ABL profile with an axisymmetric wake, it can be assumed that the requirement also holds for these cases (Bastankhah and Porté-Agel (2014); Stein and Kaltenbach (2019))."*

- Line 275: Please consider using a different title for this section. By the first look, "summary" may imply that this section is the summary of the whole manuscript.

o  We changed the title of this section to '*Summary of chapter 3.2*'

- Line 299: Please rephrase this sentence. It does not read well.

  o We rephrased the sentence, it now reads (ll.337):
  "*When the wind direction changes, the wake of an upstream turbine may pass over a downstream turbine with the consequence that Cε in the inflow changes e.g. from the wake to the ABL inflow. In such a scenario Cε changes with time.*"

- Line 29: "extend" should be replaced with "extent".

  o We corrected this

- Line 181: "be" in "This is achieved by measuring" should be replaced with "by".

  o We corrected this

- Line 136: "a axisymmetric" should be replaced with "an …".

  o We corrected this (l. 153 "*…have to be axisymmetric*")